# HOW DIFFUSION MODELS MEMORIZE

## ABSTRACT

Despite their success in image generation, diffusion models can memorize training data, raising serious privacy and copyright concerns. Although prior work has sought to characterize, detect, and mitigate memorization, the fundamental question of why and how it occurs remains unresolved. In this paper, we revisit the diffusion and denoising process and analyze latent space dynamics to address the question: *"How do diffusion models memorize?"* We show that memorization is driven by the *overestimation of training samples during early denoising*, which reduces diversity, collapses denoising trajectories, and accelerates convergence toward the memorized image. Specifically: (i) memorization cannot be explained by overfitting alone, as training loss is larger under memorization due to classifier-free guidance amplifying predictions and inducing overestimation; (ii) memorized prompts inject training images into noise predictions, forcing latent trajectories to converge and steering denoising toward their paired samples; and (iii) a decomposition of intermediate latents reveals how initial randomness is quickly suppressed and replaced by memorized content, with deviations from the theoretical denoising schedule correlating almost perfectly with memorization severity. Together, these results identify early overestimation as the central underlying mechanism of memorization in diffusion models.

## 1 INTRODUCTION

Following the successful adaptation of diffusion probabilistic models (Sohl-Dickstein et al., 2015) to image generation (Ho et al., 2020), diffusion models have become the leading framework ever since. However, despite surpassing prior state-of-the-art methods (Dhariwal & Nichol, 2021; Ramesh et al., 2022; Rombach et al., 2022; Nichol et al., 2022; Esser et al., 2024), they have also been shown to exhibit unintended memorization, reproducing training samples verbatim, even across different random seeds (Somepalli et al., 2023a; Carlini et al., 2023). This behavior raises serious privacy and copyright concerns, as it risks leaking sensitive or proprietary content (Carlini et al., 2022; Jiang et al., 2023).

To address this issue, prior work has sought to characterize memorization (van den Burg & Williams, 2021; Somepalli et al., 2023a;b; Carlini et al., 2023; Webster et al., 2023; Kadkhodaie et al., 2024; Ross et al., 2025; Jeon et al., 2025), or to detect and mitigate it by identifying common patterns associated with its occurrence (Wen et al., 2024; Ren et al., 2024; Hintersdorf et al., 2024; Jain et al., 2025). Yet these efforts stop short of providing a fundamental explanation for the phenomenon, leaving the central question unresolved: *"Why — and how — does memorization occur?"*

In this paper, we show that:

- While memorization is often attributed to overfitting, it cannot be explained by overfitting alone. In early denoising, the training loss is actually *larger* under memorization, driven by the ***overestimation*** of the training image $\mathbf{x}$ induced by classifier-free guidance (Ho & Salimans, 2021).

- Memorized prompts inject $-\mathbf{x}$ into their noise predictions, effectively steering the model to accurately predict $\mathbf{x}$ in the denoising process. With classifier-free guidance, this effect is amplified into overestimation, which diminishes latent diversity and causes denoising trajectories to converge quickly to $\mathbf{x}$.

- To formalize this phenomenon, we introduce a decomposition method for intermediate latents. Our analysis shows how initial randomness is quickly suppressed and overtaken by $\mathbf{x}$, where the

deviations from the theoretical schedule show an almost perfect correlation with memorization severity.

## 2 PRELIMINARY

Diffusion models consist of a *forward process* and a *reverse process* (Sohl-Dickstein et al., 2015; Ho et al., 2020). Given a real image $\mathbf{x} \sim q(\mathbf{x})$, where $q$ denotes the real image distribution, a *forward process* gradually adds noise to $\mathbf{x}$ over $T$ steps as

$$q(\mathbf{x}_t|\mathbf{x}_{t-1}) = \mathcal{N}(\mathbf{x}_t; \sqrt{1-\beta_t}\mathbf{x}_{t-1}, \beta_t\mathbf{I}), \tag{1}$$

where $\mathbf{x}_0 = \mathbf{x}$ and $\beta_t \in (0,1)$ is the variance schedule. Since the forward process is a fixed Markovian, sampling $\mathbf{x}_t$ at timestep $t$ can be derived in closed form as

$$q(\mathbf{x}_t|\mathbf{x}) = \mathcal{N}(\mathbf{x}_t; \sqrt{\bar{\alpha}_t}\mathbf{x}, (1-\bar{\alpha}_t)\mathbf{I}), \tag{2}$$

or equivalently, via reparameterization,

$$\mathbf{x}_t = \sqrt{\bar{\alpha}_t}\mathbf{x} + \sqrt{1-\bar{\alpha}_t}\boldsymbol{\epsilon}, \tag{3}$$

where $\alpha_t = 1 - \beta_t$, $\bar{\alpha}_t = \prod_{r=1}^{t} \alpha_r$, and $\boldsymbol{\epsilon} \sim \mathcal{N}(\mathbf{0}, \mathbf{I})$. Conversely, a *reverse process* generates $\mathbf{x}_0$ by denoising a sample $\mathbf{x}_T \sim p(\mathbf{x}_T) = \mathcal{N}(\mathbf{x}_T; \mathbf{0}, \mathbf{I})$ over $T$ steps as

$$p_\theta(\mathbf{x}_{t-1}|\mathbf{x}_t) = \mathcal{N}(\mathbf{x}_{t-1}; \boldsymbol{\mu}_\theta(\mathbf{x}_t, t), \boldsymbol{\Sigma}_\theta(\mathbf{x}_t, t)), \tag{4}$$

where each transition is modeled as a Gaussian distribution with mean $\boldsymbol{\mu}_\theta$ and variance $\boldsymbol{\Sigma}_\theta$. In practice, sampling efficiency can be improved by skipping steps (Song et al., 2021). Using Equation 3, we can formulate the estimation of $\mathbf{x}$ at timestep $t$ as

$$\hat{\mathbf{x}}_0^{(t)} = \frac{\mathbf{x}_t - \sqrt{1-\bar{\alpha}_t}\epsilon_\theta(\mathbf{x}_t)}{\sqrt{\bar{\alpha}_t}}, \tag{5}$$

where $\epsilon_\theta$ is a noise predictor trained to minimize the loss[1]

$$\mathcal{L} = ||\boldsymbol{\epsilon} - \epsilon_\theta(\mathbf{x}_t)||_2^2, \tag{6}$$

i.e., $\epsilon_\theta$ estimates the noise $\boldsymbol{\epsilon} \sim \mathcal{N}(\mathbf{0}, \mathbf{I})$ present in the noised sample $\mathbf{x}_t = \sqrt{\bar{\alpha}_t}\mathbf{x} + \sqrt{1-\bar{\alpha}_t}\boldsymbol{\epsilon}$. Based on Equations 3 and 5, $\mathbf{x}_{t-1}$ can be predicted from $\mathbf{x}_t$ as follows (Song et al., 2021):

$$\mathbf{x}_{t-1} = \sqrt{\bar{\alpha}_{t-1}}\hat{\mathbf{x}}_0^{(t)} + \sqrt{1-\bar{\alpha}_{t-1}}\epsilon_\theta(\mathbf{x}_t). \tag{7}$$

To guide the reverse process, diffusion models can be conditioned on text prompts using classifier-free guidance (Ho & Salimans, 2021), formulated as

$$\tilde{\epsilon}_\theta(\mathbf{x}_t, \mathbf{e}_c) = (1-g)\epsilon_\theta(\mathbf{x}_t, \mathbf{e}_\varnothing) + g\epsilon_\theta(\mathbf{x}_t, \mathbf{e}_c), \tag{8}$$

where $g$ is the guidance scale and $\mathbf{e}_c$ and $\mathbf{e}_\varnothing$ are CLIP (Radford et al., 2021) embeddings of text prompt $c$ and an empty string $\varnothing$, respectively. $\epsilon_\theta(\mathbf{x}_t, \mathbf{e}_\varnothing)$ and $\epsilon_\theta(\mathbf{x}_t, \mathbf{e}_c)$ are referred to as unconditional and conditional noise predictions, respectively. To perform guidance, text prompts are randomly replaced with $\varnothing$ during training, enabling the model to learn both unconditional and conditional predictions used in Equation 8. Additionally, instead of operating in the high-dimensional pixel space $\mathbb{R}^{\dim(\mathbf{x})}$, diffusion can be performed in a lower-dimensional latent space of well-trained autoencoders to reduce computational cost (Rombach et al., 2022). Note that throughout this paper, diffusion is performed in the latent space, and the notation $\mathbf{x}$ refers to latent representations rather than images in the pixel space.

## 3 HOW DIFFUSION MODELS MEMORIZE

### 3.1 EXPERIMENT SETUP

Throughout this paper, we conduct experiments with Stable Diffusion (SD) v1.4 (Rombach et al., 2022), SD v2.1 (StabilityAI, 2022), and RealisticVision (CivitAI, 2023), all using `float16` precision. Due to space constraints, we present results for SD v1.4 in the main paper and report results for

---

[1]The use of the prompt embedding is not made explicit here for simplicity.

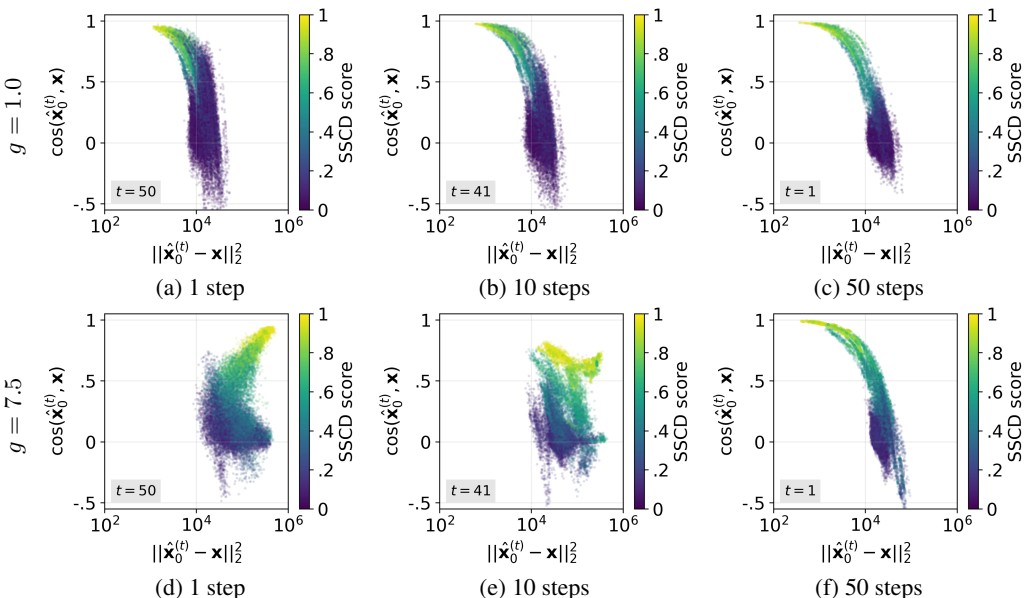

Figure 1: **Guidance amplifies the presence of x.** Squared $\ell_2$ distance (x-axis; log scale) and cosine similarity (y-axis) between $\hat{\mathbf{x}}_0^{(t)}$ and $\mathbf{x}$ after different number of denoising steps (column). The top row corresponds to $g = 1.0$, and the bottom row to $g = 7.5$. Point color denotes SSCD score. Note that the x-axis represents the *sum* of squared errors over $4 \times 64 \times 64 = 16,384$ elements (the latent dimensionality); the error in each element is very small.

the other models in Appendix F. We use DDIM (Song et al., 2021) for sampling[2], with number of inference steps $T$ as $50$ and guidance scale $g$ of $7.5$ (with classifier-free guidance) and $1.0$ (without classifier-free guidance). The dataset comprises 436 prompts from Webster (2023) (details in Appendix A). For each prompt, we generate $N = 50$ RGB images at a resolution of $512 \times 512$ pixels using distinct latents $\mathbf{x}_T \sim \mathcal{N}(\mathbf{0}, \mathbf{I})$. All computations are performed on $8\times$ NVIDIA GeForce RTX 4090 GPUs.

**Quantifying memorization.** To measure the degree of memorization in generated images, we use SSCD (Pizzi et al., 2022), which has been reported to be one of the strongest replication detectors (Somepalli et al., 2023a). Specifically, we compute two metrics: 1) $\text{SSCD}_{\text{train}}$, the similarity between a generated image $\mathbf{x}_0$ (conditioned on prompt $c$) and its paired training image $\mathbf{x}$, and 2) $\text{SSCD}_{\text{generate}}$, the mean SSCD score over all possible pairs of generated images. In our case, the average is taken across $\binom{50}{2} = 1225$ pairs. We then define the overall memorization score of a generated sample as

$$\text{SSCD score} = \frac{\text{SSCD}_{\text{train}} + \text{SSCD}_{\text{generate}}}{2}. \tag{9}$$

Unlike prior work, we introduce $\text{SSCD}_{\text{generate}}$, to account for cases where generated samples do not resemble their paired training image but remain nearly identical across different runs (Webster, 2023). In addition, we classify a generated image as memorized if SSCD score $\geq 0.75$, since scores above this threshold have been reported to indicate that two images are effectively copies of one another with $90\%$ precision (Pizzi et al., 2022).

---

[2]Our use of DDIM for sampling does not limit generality. While alternative samplers may introduce stochasticity (e.g., DDPM (Ho et al., 2020)) whereas DDIM is deterministic, our analysis remains agnostic to this distinction. Yet for completeness, we report SD v1.4 results under DDPM sampling in Appendix E.

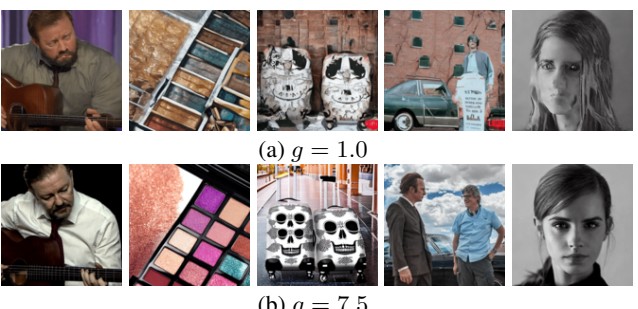

(a) $g = 1.0$

(b) $g = 7.5$

Figure 2: **Lack of guidance degrades quality.** Generated images (a) without classifier-free guidance ($g = 1.0$) and (b) with classifier-free guidance ($g = 7.5$).

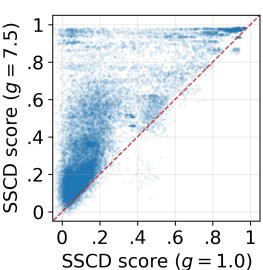

Figure 3: **Guidance drives memorization.** SSCD scores with (y-axis) and without (x-axis) classifier-free guidance.

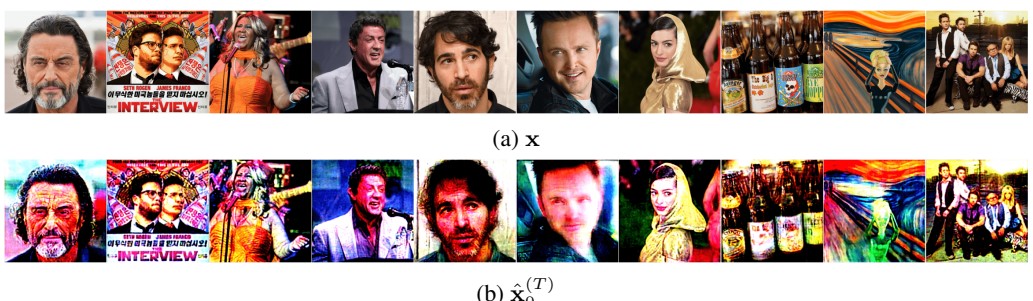

(a) $\mathbf{x}$

(b) $\hat{\mathbf{x}}_0^{(T)}$

Figure 4: **Memorization emerges from the very first step.** (a) Training images $\mathbf{x}$ and (b) their first-step predictions $\hat{\mathbf{x}}_0^{(T)}$ from paired memorized prompts $c$ (SSCD score $\geq 0.75$) under $g = 7.5$.

## 3.2 MEMORIZATION IS NOT JUST A PROBLEM OF OVERFITTING

> **Takeaway:** Memorization in diffusion models cannot be explained by overfitting alone. With classifier-free guidance, training loss is paradoxically larger in early denoising, even as memorization becomes stronger.

Although not always stated explicitly, prior work typically regards memorization as a consequence of overfitting to the training data (Section 4). We begin by examining whether this perspective holds. Note that we set $g = 1.0$ to verify whether memorization reflects overfitting, as no classifier-free guidance is applied during training. Using Equations 3 and 5, Equation 6 can be reformulated as (see Appendix D.1 for derivation):

$$\mathcal{L} = ||\frac{\sqrt{\alpha_t}}{\sqrt{1 - \bar{\alpha}_t}}(\hat{\mathbf{x}}_0^{(t)} - \mathbf{x})||_2^2. \tag{10}$$

This shows that diffusion models are trained to accurately predict $\mathbf{x}$ at every timestep (with timestep-dependent weighting). In other words, under overfitting, $||\hat{\mathbf{x}}_0^{(t)} - \mathbf{x}||_2^2 \approx 0$.

Figures 1(a–c) show $||\hat{\mathbf{x}}_0^{(t)} - \mathbf{x}||_2^2$ on the x-axis after 1, 10, and 50 denoising steps without classifier-free guidance ($g = 1.0$). This error is consistently smaller under memorization (yellow points) across all timesteps, indicating overfitting. In practice, however, classifier-free guidance is commonly used, as its absence substantially degrades generation quality (Figure 2). Will memorization still manifest as overfitting when classifier-free guidance is applied?

Figures 1(d–f) show $||\hat{\mathbf{x}}_0^{(t)} - \mathbf{x}||_2^2$ on the x-axis after 1, 10, and 50 denoising steps with classifier-free guidance applied, where $g = 7.5$. The results are striking: at earlier denoising steps, the trend *reverses*. The squared $\ell_2$ error is no longer smaller under memorization (Figure 1(e)), and at the very first step ($t = T$) it is actually *larger* (Figure 1(d)). Yet paradoxically, while appearing less like

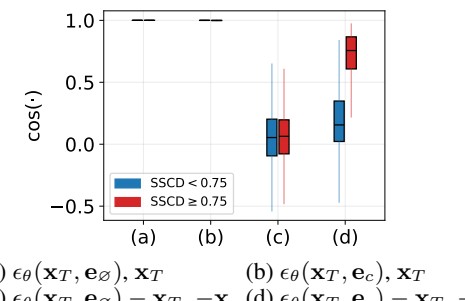

(a) $\epsilon_\theta(\mathbf{x}_T, \mathbf{e}_\varnothing), \mathbf{x}_T$      (b) $\epsilon_\theta(\mathbf{x}_T, \mathbf{e}_c), \mathbf{x}_T$
(c) $\epsilon_\theta(\mathbf{x}_T, \mathbf{e}_\varnothing) - \mathbf{x}_T, -\mathbf{x}$      (d) $\epsilon_\theta(\mathbf{x}_T, \mathbf{e}_c) - \mathbf{x}_T, -\mathbf{x}$

Figure 5: **Conditional noise prediction captures memorized data.** Cosine similarity between noise predictions and latents at $t = T$, for normal (blue; SSCD $< 0.75$) and memorized (red; SSCD $\geq 0.75$) prompts under $g = 7.5$.

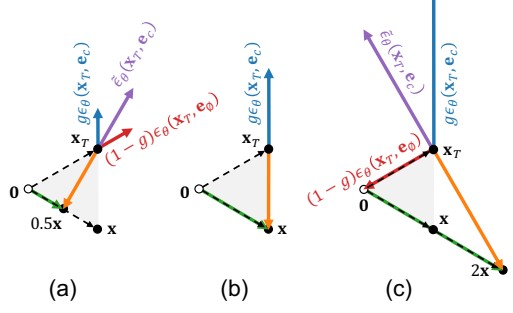

Figure 6: **Classifier-free guidance leads to overestimation of x.** Illustration of noise predictions and latents with different guidance scale $g$. (a) $g = 0.5$. (b) $g = 1.0$. (c) $g = 2.0$.

overfitting, classifier-free guidance induces stronger memorization overall (Figure 3); SSCD scores are consistently higher with guidance ($g = 7.5$, y-axis) than without it ($g = 1.0$, x-axis).

### 3.3 EARLY OVERESTIMATION ELEVATES TRAINING LOSS

> **Takeaway:** The larger training loss under classifier-free guidance arises from *overestimation of memorized data during early denoising*, driven by conditional noise predictions. This overestimation grows linearly with the guidance scale.

To understand the aforementioned discrepancy, we examine the cosine similarity between $\hat{\mathbf{x}}_0^{(t)}$ and $\mathbf{x}$ in Figures 1(a–f), plotted on the y-axis. Under memorization, the two vectors are nearly parallel across all timesteps (yellow points show consistently high cosine similarity), regardless of whether classifier-free guidance is applied. That is, $\hat{\mathbf{x}}_0^{(t)} \approx k\mathbf{x}$, with $k \approx 1$ in the absence of guidance. Under classifier-free guidance, we empirically observe $k > 1$ (see Appendix B.1). Thus, while predictions remain directionally correct in both cases, classifier-free guidance amplifies their magnitude in early denoising, leading to *overestimation of the memorized sample* $\mathbf{x}$.

The overestimation is attributed to the guidance scale $g$. Recall that $\hat{\mathbf{x}}_0^{(t)} = \frac{\mathbf{x}_t - \sqrt{1 - \bar{\alpha}_t}\tilde{\epsilon}_\theta(\mathbf{x}_t, \mathbf{e}_c)}{\sqrt{\bar{\alpha}_t}}$ (Equation 5), where $\tilde{\epsilon}_\theta(\mathbf{x}_t, \mathbf{e}_c) = (1 - g)\epsilon_\theta(\mathbf{x}_t, \mathbf{e}_\varnothing) + g\epsilon_\theta(\mathbf{x}_t, \mathbf{e}_c)$ (Equation 8). At $t = T$, three properties hold:

**A. Both unconditional and conditional noise predictions exhibit high cosine similarity with the initial latent $\mathbf{x}_T$.** Since $\epsilon = \frac{\mathbf{x}_T - \sqrt{\bar{\alpha}_T}\mathbf{x}}{\sqrt{1 - \bar{\alpha}_T}}$ ($t = T$ in Equation 3) and $\bar{\alpha}_T \approx 0$, a well-trained noise predictor $\epsilon_\theta$ will approximate $\mathbf{x}_T$ at $t = T$ ($\mathcal{L}$ sufficiently small). Figure 5(a, b) confirm this, showing that the cosine similarities between $\mathbf{x}_T$ and both unconditional and conditional noise predictions are nearly 1.

**B. Unconditional noise predictions contain no information about $\mathbf{x}$.** A random latent $\mathbf{x}_T \sim \mathcal{N}(\mathbf{0}, \mathbf{I})$ paired with an empty-string embedding $\mathbf{e}_\varnothing$ does not contain any information about the training image $\mathbf{x}$. Thus, the unconditional noise prediction $\epsilon_\theta(\mathbf{x}_T, \mathbf{e}_\varnothing)$ will contain only information about $\mathbf{x}_T$, with no information about $-\mathbf{x}$ (the negative term arises because $\epsilon = \frac{1}{\sqrt{1 - \bar{\alpha}_T}}\mathbf{x}_T + \frac{\sqrt{\bar{\alpha}_T}}{\sqrt{1 - \bar{\alpha}_T}}(-\mathbf{x})$). Figure 5(c) confirm this, showing that the cosine similarities between $\epsilon_\theta(\mathbf{x}_T, \mathbf{e}_\varnothing) - \mathbf{x}_T$ and $-\mathbf{x}$ are close to 0. We further validate this by showing that the squared magnitude of the difference between $\epsilon_\theta(\mathbf{x}_T, \mathbf{e}_\varnothing)$ and $\mathbf{x}_T$ is nearly zero (see Appendix B.2), demonstrating that unconditional noise predictions contain only information about $\mathbf{x}_T$ and none of $\mathbf{x}$.

**C. Conditional noise predictions contain substantial information about $\mathbf{x}$.** We have shown that $\hat{\mathbf{x}}_0^{(t)} \approx k\mathbf{x}$ under memorization, which implies that $\tilde{\epsilon}_\theta(\mathbf{x}_T, \mathbf{e}_c)$ must carry information about $\mathbf{x}$ (Equation 5). Figure 4 makes this more explicit: under memorization, a single denoising step yields

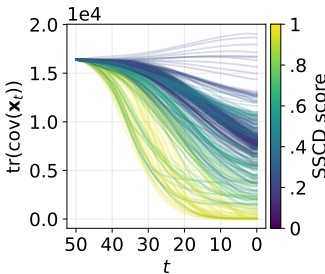

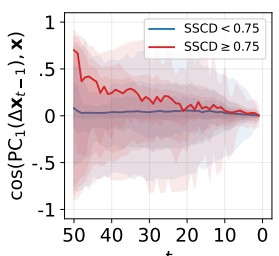

Figure 7: $\mathbf{x}_t$ **converges under early overestimation.** Trace of the covariance matrix of $\mathbf{x}_t$ as a measure of diversity across denoised latents from 50 random seeds. Colors indicate SSCD scores.

Figure 8: **Overestimation aligns denoising updates with memorized directions.** Cosine similarity between the first principal component of $\Delta\mathbf{x}_{t-1} = \mathbf{x}_{t-1} - \mathbf{x}_t$ and the memorized training image $\mathbf{x}$.

an estimate $\hat{\mathbf{x}}_0^{(T)} = \frac{\mathbf{x}_T - \sqrt{1-\bar{\alpha}_T}\tilde{\epsilon}_\theta(\mathbf{x}_T, \mathbf{e}_c)}{\sqrt{\bar{\alpha}_T}}$ that closely resembles $\mathbf{x}$. Since $\epsilon_\theta(\mathbf{x}_T, \mathbf{e}_\varnothing)$ contains no information about $\mathbf{x}$, this must be supplied by the conditional prediction $\epsilon_\theta(\mathbf{x}_T, \mathbf{e}_c)$, where the memorized prompt embedding $\mathbf{e}_c$ is provided as input (Equation 8). Figure 5(d) confirms this: although $\epsilon_\theta(\mathbf{x}_T, \mathbf{e}_c)$ is nearly (but not perfectly; see Appendix B.3) parallel to $\mathbf{x}_T$ (Figure 5(b)), unlike the unconditional case (Figure 5(c)) it also contains information about $\mathbf{x}$ (see Appendix B.2 for further verification), as $\epsilon_\theta(\mathbf{x}_T, \mathbf{e}_c) - \mathbf{x}_T$ aligns strongly with $-\mathbf{x}$ under memorization (red box plots) whereas alignment remains weak for normal prompts (blue).

Together, $\mathbf{A}$, $\mathbf{B}$, and $\mathbf{C}$ give:

$$\epsilon_\theta(\mathbf{x}_T, \mathbf{e}_\varnothing) \approx \mathbf{x}_T \text{ (from } \mathbf{A}, \mathbf{B}), \quad \epsilon_\theta(\mathbf{x}_T, \mathbf{e}_c) \approx \mathbf{x}_T - s\mathbf{x} \text{ (from } \mathbf{A}, \mathbf{C}) \tag{11}$$

for some scalar $s$. Since $\boldsymbol{\epsilon} = \frac{1}{\sqrt{1-\bar{\alpha}_T}}\mathbf{x}_T + \frac{\sqrt{\bar{\alpha}_T}}{\sqrt{1-\bar{\alpha}_T}}(-\mathbf{x})$, we infer $s \approx \frac{\sqrt{\bar{\alpha}_T}}{\sqrt{1-\bar{\alpha}_T}}$. Hence,

$$\tilde{\epsilon}_\theta(\mathbf{x}_T, \mathbf{e}_c) \approx \mathbf{x}_T - g\frac{\sqrt{\bar{\alpha}_T}}{\sqrt{1-\bar{\alpha}_T}}\mathbf{x}. \tag{12}$$

Substituting this into Equation 5 yields

$$\hat{\mathbf{x}}_0^{(T)} = \frac{\mathbf{x}_T - \sqrt{1-\bar{\alpha}_T}\tilde{\epsilon}_\theta(\mathbf{x}_T, \mathbf{e}_c)}{\sqrt{\bar{\alpha}_T}} \approx g\mathbf{x}. \tag{13}$$

Thus, *increasing the guidance scale $g$ linearly amplifies the contribution of $\mathbf{x}$ in $\hat{\mathbf{x}}_0^{(T)}$, directly causing overestimation and elevating training loss.*

Figure 6 visualizes Equation 13, linking the observations in Figure 5 to overestimation. In the figure, the origin $\mathbf{0}$ is marked by a white dot, while $\mathbf{x}_T$ and $\mathbf{x}$ are marked by black dots. Scaled noise predictions $(1-g)\epsilon_\theta(\mathbf{x}_T, \mathbf{e}_\varnothing)$, $g\epsilon_\theta(\mathbf{x}_T, \mathbf{e}_c)$, and their combination $\tilde{\epsilon}_\theta(\mathbf{x}_T, \mathbf{e}_c)$ are shown as red, blue, and purple arrows, respectively. Orange arrows denote $-\tilde{\epsilon}_\theta(\mathbf{x}_T, \mathbf{e}_c)$, with tips $(\mathbf{x}_T - \tilde{\epsilon}_\theta(\mathbf{x}_T, \mathbf{e}_c))$ pointing along the direction of $\hat{\mathbf{x}}_0^{(T)}$, shown in green arrows.

Figure 6(a) shows the case where $g = 0.5$. The green arrow points to $0.5\mathbf{x}$, yielding $\hat{\mathbf{x}}_0^{(T)} \approx 0.5\mathbf{x}$. When $g = 1.0$ (Figure 6(b)), $(1-g)\epsilon_\theta(\mathbf{x}_T, \mathbf{e}_\varnothing) = \mathbf{0}$ and $\tilde{\epsilon}_\theta(\mathbf{x}_T, \mathbf{e}_c)$ becomes $\epsilon_\theta(\mathbf{x}_T, \mathbf{e}_c)$. Thus, the green arrow points exactly $\mathbf{x}$, i.e., $\hat{\mathbf{x}}_0^{(T)} \approx \mathbf{x}$ (also shown in Figures 1(a-c)). When $g > 1$, e.g., $g = 2.0$ (Figure 6(c)), the direction of $\epsilon_\theta(\mathbf{x}_T, \mathbf{e}_\varnothing)$ flips while $\epsilon_\theta(\mathbf{x}_T, \mathbf{e}_c)$ increases in magnitude, giving $\tilde{\epsilon}_\theta(\mathbf{x}_T, \mathbf{e}_c)$ that produces $\hat{\mathbf{x}}_0^{(T)} \approx 2\mathbf{x}$.

### 3.4 WHY OVERESTIMATION IN EARLY DENOISING DRIVES MEMORIZATION

> **Takeaway:** Overestimation in early denoising acts like a "gravitational pull" toward the memorized image: it collapses latent diversity and locks trajectories onto nearly identical paths towards the memorized sample. This effect stems from excessive injection of memorized content and premature loss of randomness, with deviations from the theoretical schedule correlating almost perfectly with memorization severity.

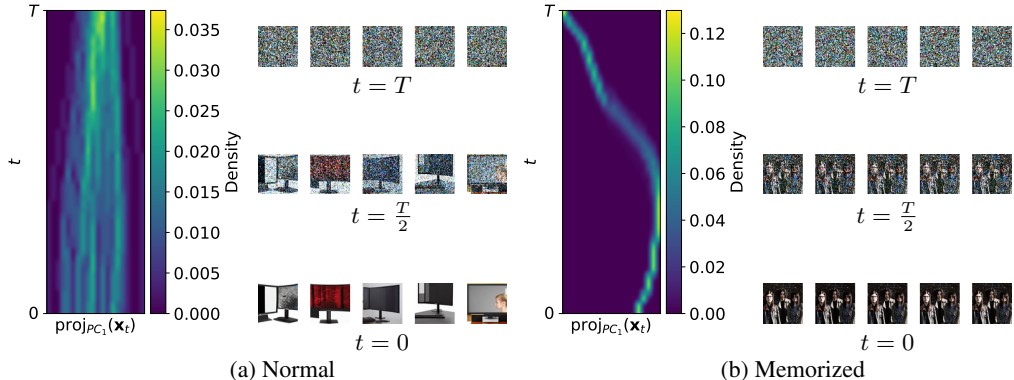

(a) Normal                                                    (b) Memorized

Figure 9: $\mathbf{x}_t$ **converges under memorization.** 1D projections of denoised latents and their decoded images at successive timesteps, where the denoising process proceeds from top to bottom (details in Appendix C). (a) Generation with a normal prompt. (b) Generation with a memorized prompt.

Then, why does overestimation in early denoising lead to severe memorization, as seen in Figure 3? To understand this, we must recall that $\epsilon_\theta$ takes only the intermediate latent $\mathbf{x}_t$ and the text embedding $\mathbf{e}_c$ (or $\mathbf{e}_\varnothing$) as inputs to predict the noise in $\mathbf{x}_t$. Since $\mathbf{e}_c$ is fixed across timesteps, timestep-dependent predictions are primarily driven by the variation in $\mathbf{x}_t$. Figure 7 plots the trace of the covariance matrix of $\mathbf{x}_t$ denoised from distinct noise samples $\mathbf{x}_T$ across timesteps, which reflects the variation of $\mathbf{x}_t$. Under memorization, the trace is smaller (yellow lines), indicating reduced diversity among denoised latents.

This reduction arises because overestimation forces each latent $\mathbf{x}_t$ to inherit a larger fraction of identical information $\mathbf{x}$ across runs, thereby suppressing variability. As a result, noise predictions under memorization become highly similar at early timesteps, pushing denoising trajectories onto nearly the same path. Figure 8 further supports this: the first principal component of $\Delta\mathbf{x}_{t-1} = \mathbf{x}_{t-1} - \mathbf{x}_t$ aligns strongly with a single direction, namely the memorized training image $\mathbf{x}$ in early denoising under memorization (red line). Consequently, while latents diverge across different $\mathbf{x}_T$ for normal prompts (Figure 9(a)), memorized prompts exhibit strong convergence: the latents are consistently pulled toward the memorized image, collapsing into nearly identical trajectories (Figure 9(b)). Thus, *early convergence of latents caused by overestimation is the key driver of memorization.* Notably, the trace in Figure 7 after only 10 denoising steps ($t = 40$) already shows strong correlation with SSCD scores (Pearson correlation coefficient $= 0.7148$).

For further investigation of convergence of $\mathbf{x}_t$ towards its destination under memorization, we introduce a decomposition method for an intermediate denoised latent $\mathbf{x}_t$ under memorization, i.e., $\mathcal{L} \approx 0$ (see Appendix D.2 for derivation):

$$\mathbf{x}_t = \sqrt{\bar{\alpha}_t}\mathbf{x} + \sqrt{1 - \bar{\alpha}_t}\mathbf{x}_T. \tag{14}$$

In other words, denoising can be interpreted as progressively suppressing the initial noise term $\mathbf{x}_T \sim \mathcal{N}(\mathbf{0}, \mathbf{I})$ while increasing the contribution of the clean latent $\mathbf{x}$. We verify Equation 14 by solving a least-squares problem $\mathbf{x}_t = w_0^{(t)}\mathbf{x} + w_T^{(t)}\mathbf{x}_T$ and comparing $w_0^{(t)}$ and $w_T^{(t)}$ to $\sqrt{\bar{\alpha}_t}$ and $\sqrt{1 - \bar{\alpha}_t}$, respectively. The results, shown in Figures 10(a, b, e, f), can be summarized as follows:

**Figures 10(a, b).** Without classifier-free guidance ($g = 1.0$), $w_0^{(t)}$ and $w_T^{(t)}$ track $\sqrt{\bar{\alpha}_t}$ and $\sqrt{1 - \bar{\alpha}_t}$ more closely under memorization (solid lines align better with dashed lines in Figure 10(b)). This is expected, as Equation 14 assumes $\mathcal{L} \approx 0$, i.e., memorization arising from overfitting. For normal prompts, $w_T^{(t)}$ roughly follows $\sqrt{1 - \bar{\alpha}_t}$, but $w_0^{(t)} \approx 0$ across timesteps (blue solid line in Figure 10(a)), indicating that the output does not resemble $\mathbf{x}$ and that other components are being constructed during denoising (as we discuss later).

**Figures 10(e, f).** Under memorization with classifier-free guidance ($g = 7.5$), the contribution of $\mathbf{x}$ is amplified in early denoising (Equation 13), leading to excessive injection of $\mathbf{x}$ into the subsequent latent (Equation 7). In other words, $w_0^{(t)}$ grows faster than its theoretical schedule $\sqrt{\bar{\alpha}_t}$, while $w_T^{(t)}$ correspondingly falls below its schedule $\sqrt{1 - \bar{\alpha}_t}$. This pattern is evident in Figure 10(f): $w_0^{(t)}$ (blue

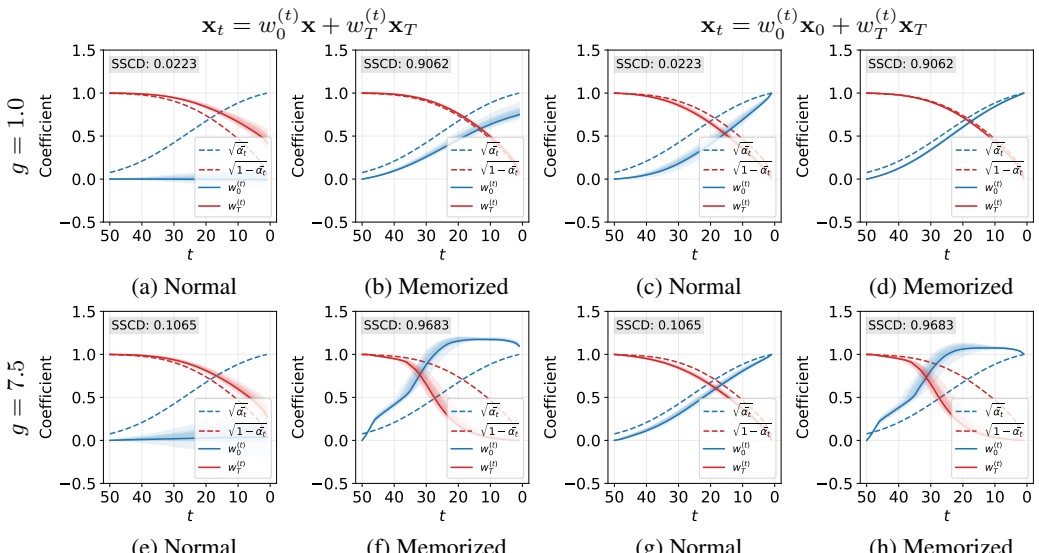

Figure 10: **Faster dominance of x under memorization.** Decomposition of $\mathbf{x}_t$ into contributions from the clean latent $\mathbf{x}$ or $\mathbf{x}_0$ ($w_0^{(t)}$; blue, solid) and initial noise $\mathbf{x}_T$ ($w_T^{(t)}$; red, solid), compared against the theoretical schedules $\sqrt{\bar{\alpha}_t}$ (blue, dashed) and $\sqrt{1-\bar{\alpha}_t}$ (red, dashed).

solid line) overshoots its theoretical curve (blue dashed line), and $w_T^{(t)}$ (red solid line) drops to zero much earlier than its theoretical curve (red dashed line). Results for normal prompts remain similar regardless of guidance (Figure 10(e)).

Even when the training image $\mathbf{x}$ is unknown, a similar trend emerges. Figures 10(c, d, g, h) present decomposition results for the least-squares problem $\mathbf{x}_t = w_0^{(t)}\mathbf{x}_0 + w_T^{(t)}\mathbf{x}_T$, where the final denoised latent $\mathbf{x}_0$ is used in place of the training image $\mathbf{x}$. For normal prompts, $w_0^{(t)}$ and $w_T^{(t)}$ generally follow their theoretical schedules (Figures 10(c, g)). This indicates that the component reinforced during denoising in Figures 10(a, b) is $\mathbf{x}_0$, meaning the destination of denoising is established early and progressively amplified throughout the process. Under memorization, the fit is nearly exact without guidance (Figure 10(d)), and with guidance $w_0^{(t)}$ grows too rapidly while $w_T^{(t)}$ decays too quickly (Figure 10(h)), once again revealing the overestimation of $\mathbf{x}$ in early denoising.

Finally, we further validate our explanation of how memorization arises by directly connecting these decompositions to SSCD scores. To this end, we compute the following three quantities, each aggregated across timesteps: 1) $\sum_{t=T}^{1} \left( \mathbb{E}[w_0^{(t)}] - \sqrt{\bar{\alpha}_t} \right)$, the *excess contribution of the memorized sample* $\mathbf{x}$. A larger value indicates that the model injects more of the training image than expected, reflecting overestimation of $\mathbf{x}$; 2) $-\sum_{t=T}^{1} \left( \mathbb{E}[w_T^{(t)}] - \sqrt{1-\bar{\alpha}_t} \right)$, the *premature suppression of the initial noise* $\mathbf{x}_T$. A large value implies that $\mathbf{x}_T$ vanishes too quickly, leaving $\mathbf{x}$ to dominate much earlier than the schedule prescribes; and 3) $\sum_{t=T}^{1} \left\{ \left( \mathbb{E}[w_0^{(t)}] - \sqrt{\bar{\alpha}_t} \right) - \left( \mathbb{E}[w_T^{(t)}] - \sqrt{1-\bar{\alpha}_t} \right) \right\}$, which reflects the *overall deviation from the theoretical denoising trajectory*.

Figures 11(a–c) plot SSCD scores against the three quantities (blue, red, and purple scatter plots, respectively). For simplicity, we omit the $\sqrt{\bar{\alpha}_t}$ and $\sqrt{1-\bar{\alpha}_t}$ terms, since they are constant across generations and do not affect comparisons. We find strong positive correlations, with Pearson coefficients of 0.9203, 0.6997, and 0.9224, respectively. These results provide direct quantitative evidence that memorization is a deterministic outcome of early overestimation: too much $\mathbf{x}$ injected too soon, and too little $\mathbf{x}_T$ left to sustain diversity.

## 4  RELATED WORK

**Memorization and overfitting.** A common view in prior work is that memorization in diffusion models arises from overfitting (Kadkhodaie et al., 2024). van den Burg & Williams (2021) showed

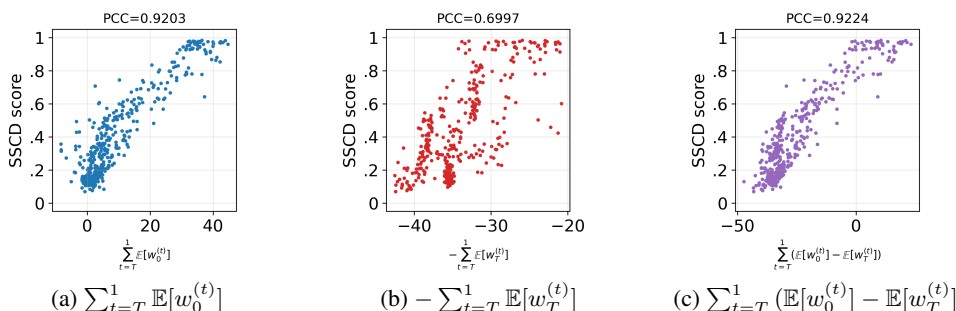

(a) $\sum_{t=T}^{1} \mathbb{E}[w_0^{(t)}]$    (b) $-\sum_{t=T}^{1} \mathbb{E}[w_T^{(t)}]$    (c) $\sum_{t=T}^{1} \left(\mathbb{E}[w_0^{(t)}] - \mathbb{E}[w_T^{(t)}]\right)$

Figure 11: **Decomposition deviations predict memorization severity.** Correlations between SSCD scores and three decomposition-based metrics.

that removing a sample from training data induces local density changes, indicating overfitting to that sample. Other studies found that duplicated images are more likely to be reproduced (Nichol, 2022; Somepalli et al., 2023a;b; Carlini et al., 2023; Webster et al., 2023). Yoon et al. (2023) showed that suppressing memorization improves generalization. Recent works propose geometric views: memorization occurs when the learned manifold contains a low-dimensional training point, yielding low variance, high sharpness, and overfitting (Ross et al., 2025). Similarly, Jeon et al. (2025) link memorization to sharp regions of the probability landscape, supporting the view that it reflects structural overfitting.

**Text guidance and memorization.** Text guidance is conjectured to be a key cause of memorization in prior work. For example, Somepalli et al. (2023b) acknowledge that text conditioning plays an important role in memorization. However, they do not analyze how text conditioning actually contributes to memorization. Similarly, Chen et al. (2024) propose a guidance strategy, Anti-Memorization Guidance (AMG), aiming at reducing three potential causes of memorization: image duplication, caption duplication, and highly specific prompts. However, they focus on proposing a mitigation strategy without analyzing how memorization is triggered or becomes severe. Our analysis provides a precise theoretical explanation: text guidance induces overestimation, linearly amplifying conditional predictions and injecting excessive information about the training image $\mathbf{x}$.

**Detection and mitigation.** Prior work explores both detection and mitigation strategies. One approach trains on intentionally corrupted data to reduce overfitting (Daras et al., 2023). Another perturbs prompts, e.g., by inserting random tokens, to discourage reproducing training images (Somepalli et al., 2023b). Detection methods include analyzing cross-attention maps (Ren et al., 2024) and localizing memorized content at the neuron level (Hintersdorf et al., 2024).

**Explanation of Wen et al. (2024).** A widely used detection method was introduced by Wen et al. (2024), which measures the magnitude of text-conditional noise predictions, i.e., $\|\epsilon_\theta(\mathbf{x}_t, \mathbf{e}_c) - \epsilon_\theta(\mathbf{x}_t, \mathbf{e}_\varnothing)\|_2$, and achieves near-perfect accuracy in identifying memorized samples. Their rationale for this choice, however, is largely heuristic, summarized as "text guidance should be larger under memorization." Our analysis provides a precise theoretical explanation: this magnitude is directly proportional to the amount of information from $\mathbf{x}$ injected at timestep $t = T$ (Equation 11). Thus, under memorization, the signal reflects the amplified contribution of the memorized data $\mathbf{x}$ at every denoising step, establishing it as a principled and reliable metric for detecting memorization.

**Explanation of Jain et al. (2025).** A recent work proposed mitigation by identifying a transition timestep in denoising (Jain et al., 2025): classifier-free guidance is disabled before that timestep and enabled afterward, which prevents memorized generations. However, this strategy is based on empirical observations, without a clear explanation of why it works. Our analysis clarifies the mechanism: early denoising is precisely where classifier-free guidance induces overestimation, linearly amplifying conditional predictions and injecting excessive information about the training image $\mathbf{x}$. By withholding guidance during these steps, latents retain randomness and spread into diverse, non-memorized directions. Once sufficient diversity and stable trajectories are established, guidance can be safely reintroduced.

## 5 CONCLUSION

In this paper, we revisited the denoising dynamics of diffusion models to answer the question: *"how do they memorize?"* We showed that memorization is not simply an artifact of overfitting during training, but arises from *overestimation of memorized data in early denoising*, where classifier-free guidance linearly amplifies conditional predictions and injects too much of the training image too soon. This amplification collapses latent diversity and locks trajectories onto nearly identical paths, rapidly erasing randomness and replacing it with the memorized content. We believe that recognizing and shaping this regime would provide a practical path toward safer, less replicative generative systems.

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

# A DATASET

In this paper, we use prompts from Webster (2023), all sourced from the LAION-5B dataset (Schuhmann et al., 2022) used to train the diffusion models studied here, namely, SD v1.4 (Rombach et al., 2022), SD v2.1 (StabilityAI, 2022), and RealisticVision (CivitAI, 2023). The prompts are grouped into four categories:

**(i) Matching verbatim (MV)**: pixel-level memorization, where the generated output exactly reproduces a training image;

**(ii) Template duplicate (TV)**: images that share the overall template with a training image but differ in details such as colors or textures;

**(iii) Retrieved verbatim (RV)**: cases where the output does not match the paired training image but consistently reproduces another image from the training set across different runs; and

**(iv) None (N)**: normal prompts that produce diverse outputs across runs without reproducing training images.

Each prompt was provided with a URL linking to its paired training image. However, some URLs were inaccessible, preventing retrieval of the corresponding training images. We exclude such cases and use only prompts with retrievable images, as shown in Table 1. Furthermore, we observed that certain prompts were miscategorized in the original groupings. For instance, one prompt labeled as N produced pixel-level memorized images (MV). To address this, we discard the categorizations of Webster (2023) and instead re-score prompts using SSCD (Pizzi et al., 2022) (Section 3.1), which is then used for all subsequent analyses.

Table 1: Prompt categories and counts across different diffusion models. Fractions indicate the number of prompts with retrievable paired training images over the total number of prompts originally provided by Webster (2023).

|  | MV | TV | RV | N | Total |
|---|---|---|---|---|---|
| SD v1.4 | 74/86 | 208/229 | 30/30 | 124/155 | 436/500 |
| SD v2.1 | 3/4 | 198/215 | 0/0 | 188/281 | 389/500 |
| RealisticVision | 78/90 | 209/230 | 34/34 | 114/146 | 435/500 |

# B ADDITIONAL EVIDENCES

## B.1 ADDITIONAL EVIDENCE OF OVERESTIMATION

Figure 12 shows that under memorization $k = \frac{||\hat{\mathbf{x}}_0^{(t)}||_2}{||\mathbf{x}||_2} > 1$, with red regions lying to the right of the dashed vertical line at $k = 1$. This provides clear evidence of overestimation rather than underestimation of $\mathbf{x}$.

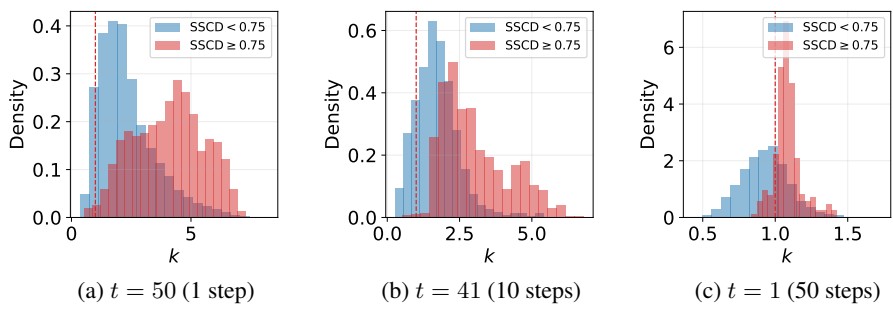

(a) $t = 50$ (1 step)    (b) $t = 41$ (10 steps)    (c) $t = 1$ (50 steps)

Figure 12: **Overestimation occurs under memorization.** Distribution of $k = \frac{||\hat{\mathbf{x}}_0^{(t)}||_2}{||\mathbf{x}||_2}$ across timesteps $t$. The red dashed line marks $k = 1$.

## B.2    ADDITIONAL EVIDENCE FOR FIGURE 5(C, D)

Figure 13(a) shows the squared magnitude of the difference between $\epsilon_\theta(\mathbf{x}_T, \mathbf{e}_\varnothing)$ and $\mathbf{x}_T$ at $t = T$. The difference is nearly zero, confirming that unconditional noise predictions reproduce $\mathbf{x}_T$ and contain no information about $\mathbf{x}$.

Figure 13(b) reports the squared magnitude of the difference between $\epsilon_\theta(\mathbf{x}_T, \mathbf{e}_c)$ and $\mathbf{x}_T$ at $t = T$ for normal (blue; SSCD score $< 0.75$) and memorized (red; SSCD score $\geq 0.75$) prompts. For normal prompts, the distribution closely matches the unconditional case (Figure 13(a)). Under memorization, however, the distribution shifts to larger values, indicating that conditional predictions contain information beyond $\mathbf{x}_T$, specifically the contribution of $-\mathbf{x}$ (Figure 5(d)).

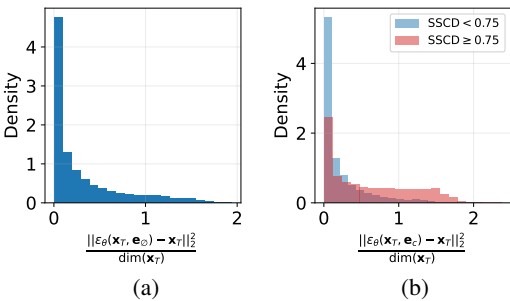

(a)                                    (b)

Figure 13: **Unconditional predictions replicate $\mathbf{x}_T$, whereas conditional predictions also contain memorized information.** Distribution of the squared magnitude of the difference between $\mathbf{x}_T$ and (a) unconditional noise predictions and (b) conditional noise predictions at $t = T$.

## B.3    ADDITIONAL EXPLANATION FOR FIGURE 5(B, D)

Figure 14 illustrates how $\epsilon_\theta(\mathbf{x}_T, \mathbf{e}_c)$ can contain information about $\mathbf{x}$ even when its cosine similarity with $\mathbf{x}_T$ is nearly 1 (Figure 5(b)). The median of cosine similarity between $\epsilon_\theta(\mathbf{x}_T, \mathbf{e}_c)$ and $\mathbf{x}$ for memorized samples is $0.7543$ (Figure 5(d)), corresponding to an angle of roughly $\arccos(0.7543) \approx 41.07°$. As shown in Figure 14, $\epsilon_\theta(\mathbf{x}_T, \mathbf{e}_c)$ (blue arrow) and $\mathbf{x}_T$ (black arrow) appear nearly parallel ($\approx 0°$ apart), but a residual component remains between them (green arrow; $\epsilon_\theta(\mathbf{x}_T, \mathbf{e}_c) - \mathbf{x}_T$), namely, $-s\mathbf{x}$. This geometric gap demonstrates how conditional predictions contain additional information about $\mathbf{x}$, despite strong alignment with $\mathbf{x}_T$.

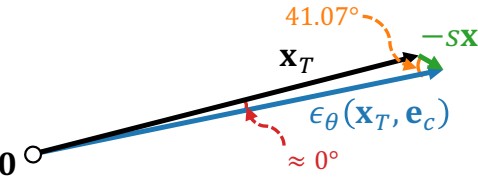

Figure 14: **Conditional noise predictions contain information about $\mathbf{x}$ even when nearly parallel to $\mathbf{x}_T$.** Geometric illustration showing $\epsilon_\theta(\mathbf{x}_T, \mathbf{e}_c)$ (blue), $\mathbf{x}_T$ (black), and $-s\mathbf{x}$ (green).

## C    NOTES ON FIGURE 9

Figure 9 visualizes the evolution of latents during denoising by projecting all $N \times (T + 1) = 50 \times 51 = 2550$ latents for a given prompt onto their first principal component from PCA. The term $T + 1$ arises because we include the initial random latents $\mathbf{x}_T$ along with the $T$ subsequent denoised states.

## D PROOFS

### D.1 EQUATION 10

The diffusion training loss is originally defined as Equation 6:

$$\mathcal{L} = ||\boldsymbol{\epsilon} - \epsilon_\theta(\mathbf{x}_t, \mathbf{e}_c)||_2^2. \tag{15}$$

From Equation 3, the ground-truth noise $\boldsymbol{\epsilon}$ can be written as

$$\boldsymbol{\epsilon} = \frac{\mathbf{x}_t - \sqrt{\bar{\alpha}_t}\mathbf{x}}{\sqrt{1 - \bar{\alpha}_t}}, \tag{16}$$

and from Equation 5, the predicted noise $\epsilon_\theta(\mathbf{x}_t, \mathbf{e}_c)$ is

$$\epsilon_\theta(\mathbf{x}_t, \mathbf{e}_c) = \frac{\mathbf{x}_t - \sqrt{\bar{\alpha}_t}\hat{\mathbf{x}}_0^{(t)}}{\sqrt{1 - \bar{\alpha}_t}}. \tag{17}$$

Substituting Equations 16 and 17 into Equation 15 yields

$$\mathcal{L} = ||\frac{\sqrt{\bar{\alpha}_t}}{\sqrt{1 - \bar{\alpha}_t}}(\hat{\mathbf{x}}_0^{(t)} - \mathbf{x})||_2^2. \blacksquare \tag{18}$$

### D.2 EQUATION 14

Under memorization, $\mathcal{L} \approx 0$, which is equivalent to $\boldsymbol{\epsilon} \approx \epsilon_\theta(\mathbf{x}_t, \mathbf{e}_c)$ ($\because$ Equation 6). However, $\boldsymbol{\epsilon}$ is independent of $t$, thus we can write

$$\boldsymbol{\epsilon} = \frac{\mathbf{x}_T - \sqrt{\bar{\alpha}_T}\mathbf{x}}{\sqrt{1 - \bar{\alpha}_T}} \approx \epsilon_\theta(\mathbf{x}_t, \mathbf{e}_c). \tag{19}$$

Therefore, $\mathbf{x} \approx \hat{\mathbf{x}}_0^{(t)} \approx \frac{\mathbf{x}_t - \sqrt{1 - \bar{\alpha}_t}\boldsymbol{\epsilon}}{\sqrt{\bar{\alpha}_t}}$ ($\because$ Equations 5 and 10). In other words,

$$\mathbf{x}_t = (\sqrt{\bar{\alpha}_t} - \frac{\sqrt{\bar{\alpha}_T}}{\sqrt{1 - \bar{\alpha}_T}})\mathbf{x} + \frac{\sqrt{1 - \bar{\alpha}_t}}{\sqrt{1 - \bar{\alpha}_T}}\mathbf{x}_T, \tag{20}$$

or using $\bar{\alpha}_T \approx 0$,

$$\mathbf{x}_t = \sqrt{\bar{\alpha}_t}\mathbf{x} + \sqrt{1 - \bar{\alpha}_t}\mathbf{x}_T. \blacksquare \tag{21}$$

## E DDPM

In this section, we demonstrate that our findings are not tied to a specific sampling method. In particular, we obtain consistent results under DDPM sampling (Ho et al., 2020), which introduces stochasticity. Note that we use $N = 10$.

With DDPM, we again observe overfitting under memorization without classifier-free guidance (Figures 15(a–c)) and early overestimation with classifier-free guidance (Figures 15(d–f)), mirroring the trends seen with DDIM sampling.

We also derive an analogous decomposition for DDPM and confirm that the results remain unchanged. The DDPM reverse transition (Ho et al., 2020) is

$$\mathbf{x}_{t-1} = \frac{1}{\sqrt{\alpha_t}}\Big(\mathbf{x}_t - \frac{\beta_t}{\sqrt{1 - \bar{\alpha}_t}}\epsilon_\theta(\mathbf{x}_t, \mathbf{e}_c)\Big) + \sigma_t \mathbf{z}, \qquad \mathbf{z} \sim \mathcal{N}(\mathbf{0}, \mathbf{I}), \tag{22}$$

where

$$\sigma_t^2 = \tilde{\beta}_t = \frac{1 - \bar{\alpha}_{t-1}}{1 - \bar{\alpha}_t}\beta_t. \tag{23}$$

Under memorization, we again have $\epsilon_\theta(\mathbf{x}_t, \mathbf{e}_c) \approx \boldsymbol{\epsilon}$ (by the same reasoning as in Equation 19). Substitute this into Equation 22 and using the forward-process identity $\mathbf{x}_t = \sqrt{\bar{\alpha}_t}\,\mathbf{x} + \sqrt{1 - \bar{\alpha}_t}\,\boldsymbol{\epsilon}$

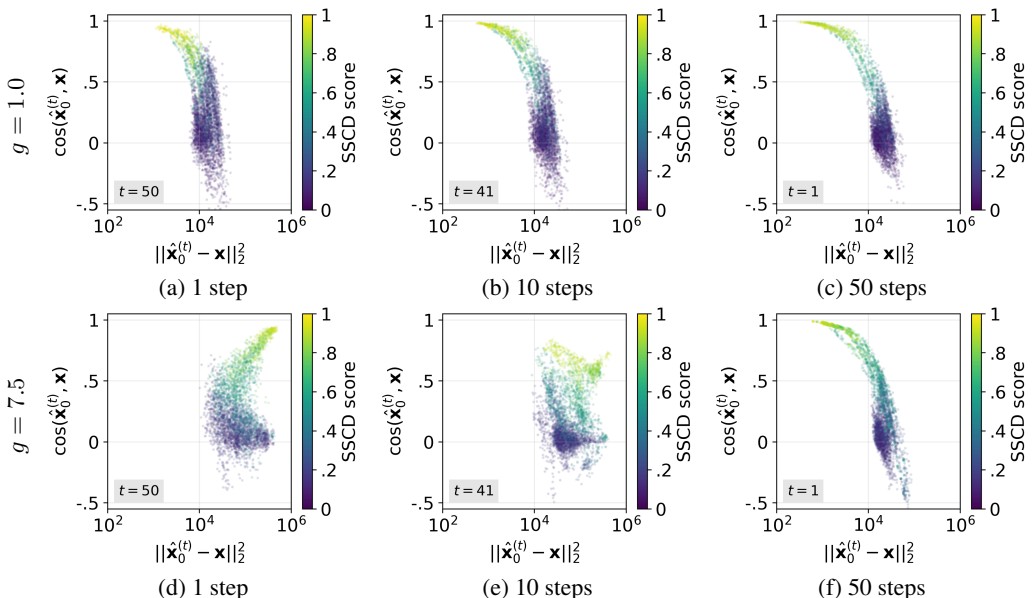

Figure 15: **Guidance amplifies the presence of x.** Squared $\ell_2$ distance (x-axis; log scale) and cosine similarity (y-axis) between $\hat{\mathbf{x}}_0^{(t)}$ and $\mathbf{x}$ after different number of denoising steps (column). The top row corresponds to $g = 1.0$, and the bottom row to $g = 7.5$. Point color denotes SSCD score.

(Equation 3) gives

$$\mathbf{x}_{t-1} \approx \frac{1}{\sqrt{\alpha_t}}\left(\sqrt{\bar{\alpha}_t}\,\mathbf{x} + \sqrt{1-\bar{\alpha}_t}\,\boldsymbol{\epsilon} - \frac{\beta_t}{\sqrt{1-\bar{\alpha}_t}}\,\boldsymbol{\epsilon}\right) + \sigma_t\,\mathbf{z}$$

$$= \underbrace{\frac{\sqrt{\bar{\alpha}_t}}{\sqrt{\alpha_t}}}_{=\sqrt{\bar{\alpha}_{t-1}}}\mathbf{x} + \frac{1}{\sqrt{\alpha_t}}\frac{(1-\bar{\alpha}_t) - \beta_t}{\sqrt{1-\bar{\alpha}_t}}\,\boldsymbol{\epsilon} + \sigma_t\,\mathbf{z}. \tag{24}$$

Using the identity $(1 - \bar{\alpha}_t) - \beta_t = \alpha_t(1 - \bar{\alpha}_{t-1})$, we obtain the compact form

$$\mathbf{x}_{t-1} \approx \sqrt{\bar{\alpha}_{t-1}}\,\mathbf{x} + \frac{\sqrt{\alpha_t}\,(1-\bar{\alpha}_{t-1})}{\sqrt{1-\bar{\alpha}_t}}\,\boldsymbol{\epsilon} + \sigma_t\,\mathbf{z}. \tag{25}$$

Finally, with $\bar{\alpha}_T \approx 0$,

$$\boldsymbol{\epsilon} = \frac{\mathbf{x}_T - \sqrt{\bar{\alpha}_T}\,\mathbf{x}}{\sqrt{1-\bar{\alpha}_T}} \approx \mathbf{x}_T, \tag{26}$$

so the DDPM step under memorization decomposes as

$$\mathbf{x}_{t-1} \approx \sqrt{\bar{\alpha}_{t-1}}\,\mathbf{x} + \frac{\sqrt{\alpha_t}\,(1-\bar{\alpha}_{t-1})}{\sqrt{1-\bar{\alpha}_t}}\,\mathbf{x}_T + \sigma_t\,\mathbf{z}. \ \blacksquare \tag{27}$$

Thus, regardless of the stochasticity introduced, an intermediate latent can still be decomposed into the target image $\mathbf{x}$ and the initial random latent $\mathbf{x}_T$, with the added stochasticity appearing as an independent term. As a result, we obtain the same findings as in Figure 11: decomposition deviations remain strongly correlated with memorization severity under DDPM sampling (Figure 16), confirming that our analysis is not tied to a particular sampler.

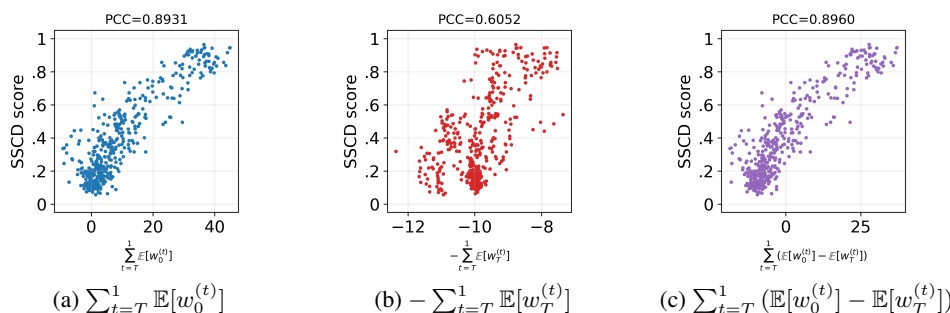

(a) $\sum_{t=T}^{1} \mathbb{E}[w_0^{(t)}]$  (b) $-\sum_{t=T}^{1} \mathbb{E}[w_T^{(t)}]$  (c) $\sum_{t=T}^{1} \left( \mathbb{E}[w_0^{(t)}] - \mathbb{E}[w_T^{(t)}] \right)$

Figure 16: **Decomposition deviations predict memorization severity.** Correlations between SSCD scores and three decomposition-based metrics.

## F  RESULTS ON OTHER MODELS

In this section, we present results for SD v2.1 (StabilityAI, 2022) and RealisticVision (CivitAI, 2023), using $N = 10$. The outcomes closely mirror those reported for SD v1.4 in the main paper, confirming that our findings hold consistently across different diffusion models[3].

---

[3]SD v2.1 exhibits substantially less memorization because of de-duplication in its training set (Nichol, 2022), leaving relatively few memorized prompts in Webster (2023) (Table 1). As a result, some quantitative values are lower, but the overall patterns and trends remain consistent and strongly support our conclusions.

## F.1 SD v2.1

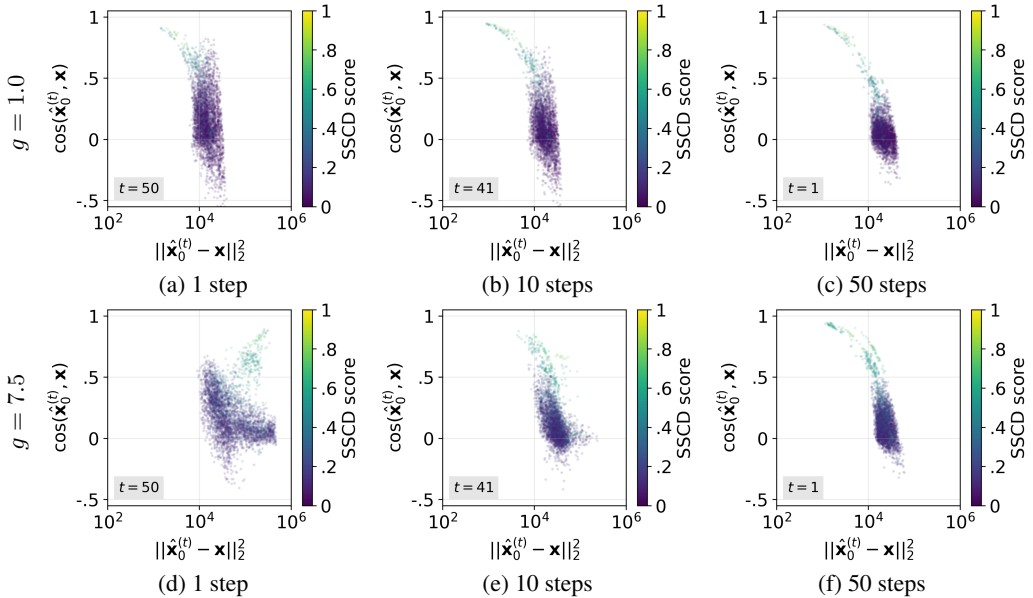

(a) 1 step
(b) 10 steps
(c) 50 steps

(d) 1 step
(e) 10 steps
(f) 50 steps

Figure 17: **Guidance amplifies the presence of x.** Squared $\ell_2$ distance (x-axis; log scale) and cosine similarity (y-axis) between $\hat{\mathbf{x}}_0^{(t)}$ and $\mathbf{x}$ after different number of denoising steps (column). The top row corresponds to $g = 1.0$, and the bottom row to $g = 7.5$. Point color denotes SSCD score.

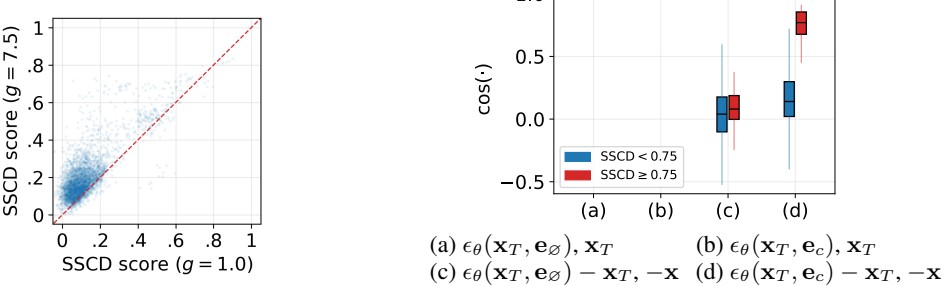

Figure 18: **Guidance drives memorization.** SSCD scores with (y-axis) and without (x-axis) classifier-free guidance.

Figure 19: **Conditional noise prediction captures memorized data.** Cosine similarity between noise predictions and latents at $t = T$, for normal (blue; SSCD < 0.75) and memorized (red; SSCD $\geq$ 0.75) prompts under $g = 7.5$.

(a) $\epsilon_\theta(\mathbf{x}_T, \mathbf{e}_\varnothing), \mathbf{x}_T$  (b) $\epsilon_\theta(\mathbf{x}_T, \mathbf{e}_c), \mathbf{x}_T$
(c) $\epsilon_\theta(\mathbf{x}_T, \mathbf{e}_\varnothing) - \mathbf{x}_T, -\mathbf{x}$  (d) $\epsilon_\theta(\mathbf{x}_T, \mathbf{e}_c) - \mathbf{x}_T, -\mathbf{x}$

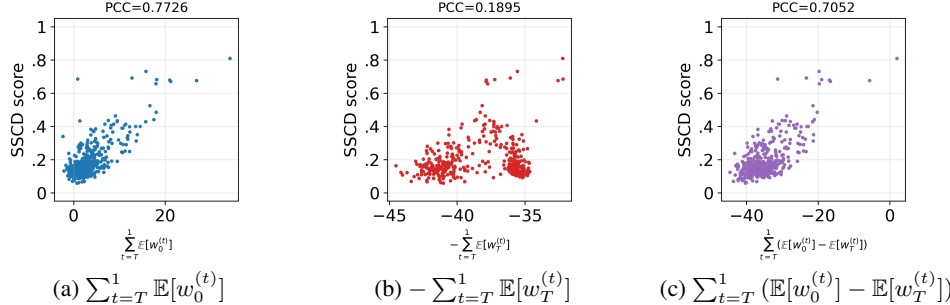

(a) $\sum_{t=T}^{1} \mathbb{E}[w_0^{(t)}]$
(b) $-\sum_{t=T}^{1} \mathbb{E}[w_T^{(t)}]$
(c) $\sum_{t=T}^{1} \left( \mathbb{E}[w_0^{(t)}] - \mathbb{E}[w_T^{(t)}] \right)$

Figure 20: **Decomposition deviations predict memorization severity.** Correlations between SSCD scores and three decomposition-based metrics.

## F.2 REALISTICVISION

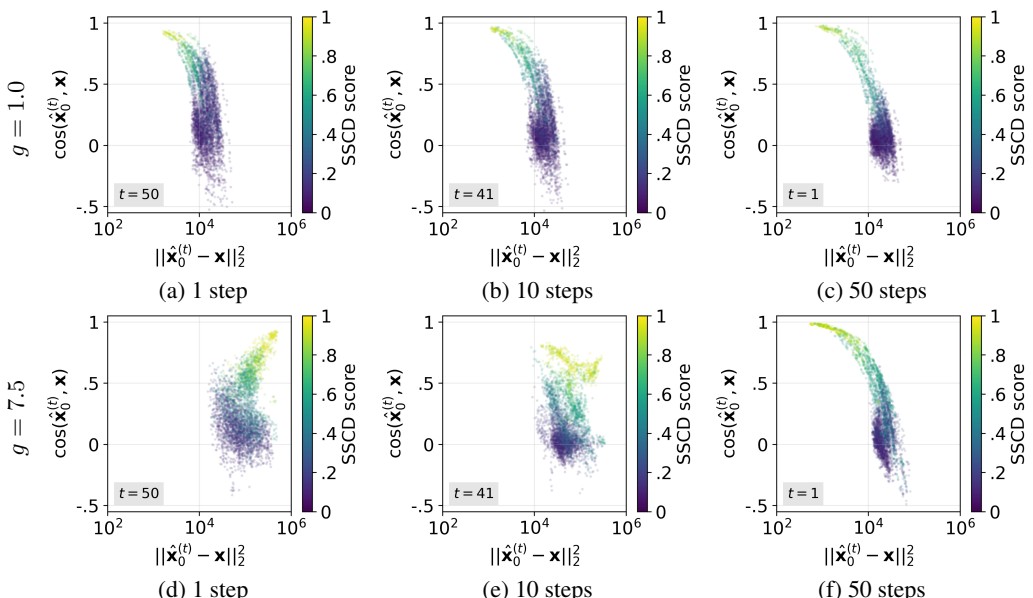

Figure 21: **Guidance amplifies the presence of x.** Squared $\ell_2$ distance (x-axis; log scale) and cosine similarity (y-axis) between $\hat{\mathbf{x}}_0^{(t)}$ and $\mathbf{x}$ after different number of denoising steps (column). The top row corresponds to $g = 1.0$, and the bottom row to $g = 7.5$. Point color denotes SSCD score.

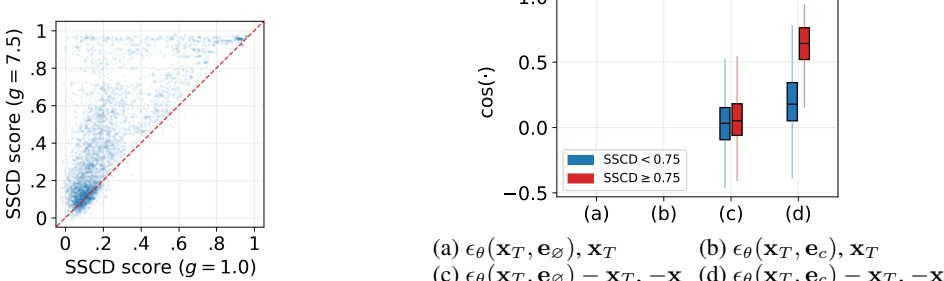

Figure 22: **Guidance drives memorization.** SSCD scores with (y-axis) and without (x-axis) classifier-free guidance.

Figure 23: **Conditional noise prediction captures memorized data.** Cosine similarity between noise predictions and latents at $t = T$, for normal (blue; SSCD $< 0.75$) and memorized (red; SSCD $\geq 0.75$) prompts under $g = 7.5$.

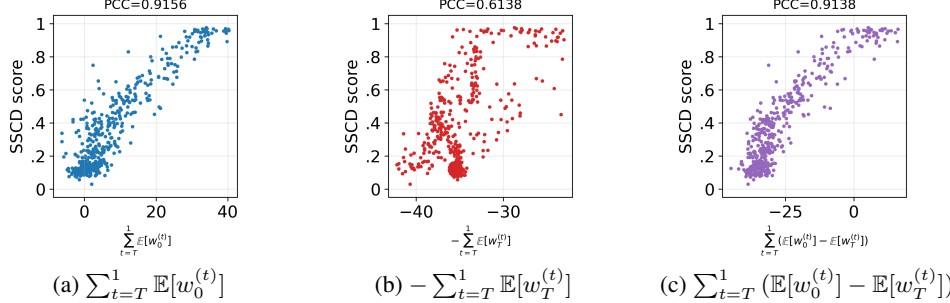

Figure 24: **Decomposition deviations predict memorization severity.** Correlations between SSCD scores and three decomposition-based metrics.

## G RESOURCE DETAILS

With data-distributed parallel (DDP) computing, logging intermediate latents and noise predictions at every denoising step for all models and schedulers used in this paper can be completed in under a day and requires roughly 6 TB of storage.

## H DETECTION AND MITIGATION OF MEMORIZATION

In this section, we present a simple mitigation strategy for memorization based on the findings of this paper. We showed that memorization is driven by the overestimation of memorized data in the early denoising steps. Therefore, a straightforward way to mitigate memorization is to reduce this overestimation when it occurs.

In Figure 11(c), we show that the decomposition deviations are strongly correlated with memorization severity. We further observe that this strong correlation is preserved at the level of individual denoising steps, i.e., $\delta^{(t)} = (w_0^{(t)} - \sqrt{\bar{\alpha}_t}) - (w_T^{(t)} - \sqrt{1 - \bar{\alpha}_t})$ is also strongly correlated with memorization severity. We leverage this property to detect memorization in real time during denoising. That is, if $\delta^{(t)} > 0$, we consider memorization has occurred at the denoising step $t$. To account for estimation noise in practice, we introduce a margin $\gamma > 0$ and consider memorization to have occurred when $\delta^{(t)} > -\gamma$. Using this method, we observe that memorization can be detected before $t = 3.31 \pm 3.69$ steps.

When memorization is detected, we set the guidance scale $g$ as $0$, i.e., we use the unconditional noise prediction $\epsilon_\theta(\mathbf{x}_t, \mathbf{e}_\varnothing)$ to predict the noise at step $t$. This prevents overestimation, as the overestimation arises specifically from predicting noise conditioned on the memorized prompt $c$. We observe that this simple intervention effectively mitigates memorization, as shown in Figure 25, further supporting the findings of this paper. Additionally, for memorized samples (SSCD score $\geq 0.75$), we observe a substantial drop in SSCD$_{\text{train}}$ (similarity between generated images and their paired training images), from $0.9203 \pm 0.0633$ to $0.1766 \pm 0.2519$, indicating that the mitigation significantly reduces replication of training images.

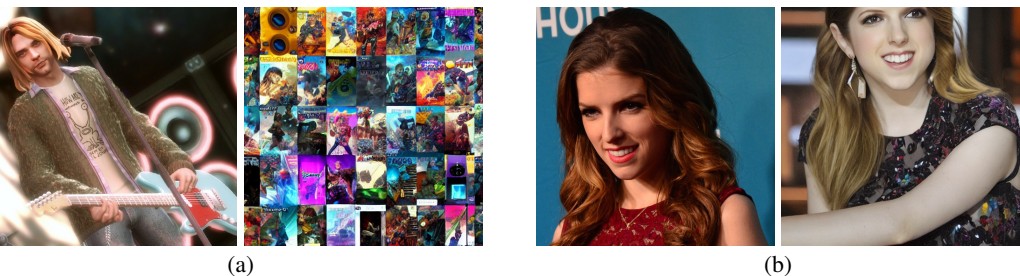

(a)                                    (b)

Figure 25: **Mitigation examples.** Memorized (left) and mitigated (right) images with $\gamma = 0.2$. Prompt: (a) *33 Screenshots of Musicians in Videogames* (b) *Anna Kendrick is Writing a Collection of Funny, Personal Essays*

## I FUTURE WORK

There are several promising directions for extending this study. One is to examine whether the early-step overestimation we observed in image diffusion models also shows up in other modalities, such as audio, video, or text. If similar patterns emerge, it would help clarify how general this mechanism is and whether different modalities exhibit their own characteristic forms of overestimation. Another direction is to study how the mechanism behaves when classifier-free guidance is weak or

absent. Our current analysis focuses on guided sampling, but unguided models or models fine-tuned with methods like LoRA may have different dynamics. Understanding whether overestimation still appears, or whether it changes in strength or form, will be interesting.

