# OpenReview forum: "How Diffusion Models Memorize"
_ICLR.cc/2026/Conference — Submitted to ICLR 2026_

### Official Review · Reviewer_B5KX · 2025-10-27

**Soundness:** 2
**Presentation:** 3
**Contribution:** 2
**Rating:** 4
**Confidence:** 4

**Summary:**

This paper investigates why diffusion models memorize training images and identifies the early overestimation of the clean x when using classifier-free guidance. The authors show that classifier-free guidance amplifies the contribution of x, leading to memorization, and that this phenomenon arises from inference-time dynamics rather than training overfitting.

**Strengths:**

Provides a clear mechanistic understanding of memorization under classifier-free guidance, distinguishing it from traditional overfitting during training.

**Weaknesses:**

I think this paper mainly analyzes how classifier-free guidance amplifies memorization, rather than explaining how diffusion models memorize in the first place. Some experiments actually meet expectations. For instance, the diffusion model itself is not trained in the same way as the CFG inference — it learns to predict noise, either conditional or unconditional, not to generate the noise after the CFG amplification — so the fact that CFG increases loss is expected. The main issue, in my view, is that even after following the authors’ reasoning, I still don’t clearly understand how diffusion models memorize at all. The paper convincingly shows that conditional guidance can trigger memorization, but it stops short of explaining the fundamental memorization mechanism inherent to the diffusion process itself.

**Questions:**

See Weakness. Further:
   1. Have you examined whether unconditional diffusion models also memorize training samples？ In 3.3 you say B: Unconditional noise predictions contain no information about x, will it contain information about other training sample?
  2. What factors explain why some prompts trigger memorization while others do not ?

---

> ### Author Response · Authors · 2025-11-21
>
> > **(Weakness)** I think this paper mainly analyzes how classifier-free guidance amplifies memorization, rather than explaining how diffusion models memorize in the first place. Some experiments actually meet expectations. For instance, the diffusion model itself is not trained in the same way as the CFG inference — it learns to predict noise, either conditional or unconditional, not to generate the noise after the CFG amplification — so the fact that CFG increases loss is expected. The main issue, in my view, is that even after following the authors’ reasoning, I still don’t clearly understand how diffusion models memorize at all. The paper convincingly shows that conditional guidance can trigger memorization, but it stops short of explaining the fundamental memorization mechanism inherent to the diffusion process itself.
>
> Thank you for raising this thoughtful concern.
> We appreciate the chance to clarify our contribution, as it seems that a few key points in our explanation may not have been fully conveyed.
>
> For memorization to occur, that is, for a training image to reappear at inference, the diffusion model must overfit to that particular image during training.
> This is known from prior work, as summarized in the first paragraph (**Memorization and overfitting.**) in Section 4.
> However, overfitting alone is insufficient to explain memorization, and addressing this gap is an important contribution of our work.
>
> Building on the fact that overfitting is the first condition for memorization, we show that memorization emerges through overestimation of an accurately predicted memorized sample (Figure 1).
> CFG then linearly amplifies this overestimation (Figure 6, Equation 13), which rapidly removes the initial randomness and drives the latent toward the memorized image (Figure 10).
> In our paper, we reveal this mechanism rigorously, which prior work has not explored.
>
>
> > **(Question 1)** Have you examined whether unconditional diffusion models also memorize training samples? In 3.3 you say B: Unconditional noise predictions contain no information about x, will it contain information about other training sample?
>
> Thank you for the interesting question, yet answering it is quite challenging.
> Without any guidance, reaching a specific memorized sample in the latent space is extremely unlikely.
> While a random initialization might occasionally land near a memorized sample, this would depend almost entirely on chance (initial randomness) rather than through any systematic behavior of the model.
> In other words, detecting memorization in unconditional models would require generating a very large number of outputs and hoping that one among them happens to match a training example, which is practically impossible.
> Therefore, we did not examine whether unconditional models memorize training samples.
> Additionally, [1] also reports that memorization is unlikely to occur in unconditional models.
>
>
> > **(Question 2)** What factors explain why some prompts trigger memorization while others do not ?
>
> As discussed in the first paragraph of Section 4, the most common cause of memorization in diffusion models is the presence of duplicates in the training set [1, 2, 3, 4, 5].
> When duplicates exist, the model assigns disproportionately high emphasis to fitting those samples, leading to overfitting.
> Consequently, prompts corresponding to such duplicated data tend to reproduce the memorized image with little to no variation.
>
> [1] Understanding and Mitigating Copying in Diffusion Models, Somepalli et al., NIPS 2023 \
> [2] Diffusion, Art or Digital Forgery? Investigating Data Replication in Diffusion Models, Somepalli et al., CVPR 2023 \
> [3] https://openai.com/index/dall-e-2-pre-training-mitigations/ \
> [4] Extracting Training Data from Diffusion Models, Carlini et al., USENIX 2023 \
> [5] On the De-duplication of LAION-2B, Webster et al., arXiv 2023

---

### Official Review · Reviewer_Xz4q · 2025-10-30

**Soundness:** 2
**Presentation:** 2
**Contribution:** 1
**Rating:** 2
**Confidence:** 4

**Summary:**

This paper studies the memorization of diffusion models. Specifically, the authors empirically show that classifier-free guidance (CFG) amplifies and strengthens memorization in diffusion models. Then, they attribute the effect of CFG to the overestimation of $x$ and show the correlation between the trajectory deviation and the memorization.

**Strengths:**

* The results and visualization are clear and convincing. They conduct a lot of experiments in Section 3.3 to validate their claims. They calculate the cosine similarity between noise predictions and the initial noise $x_T$ or the real image $x$, showing that the conditional noise predictor memorizes the training sample $x$, while the unconditional predictor does not. Then these findings are further used to formally explain the effect of CFG in Eqs. (11)-(13).

**Weaknesses:**

* My biggest concern is your claim that overfitting cannot explain memorization alone. For the CFG, if the conditional diffusion does not overfit, then CFG won’t amplify the effect. So, the memorization is still attributed to overfitting. In other words, the overfitting of conditional predictor is the necessary condition, and CFG just gives it a push.
* The preliminary is too long, introducing some basic concepts. Put some part in the appendix and only keep the most important part in your main paper, like the noise of the CFG process.
* In Section 3.2, you use the term "training loss". Are you using ||\hat_{x}_0, x_0|| as the “training loss”? To use a CFG method, we need to train conditional and unconditional diffusion models. Then, the training loss should just be the loss for training the conditional and unconditional models. But I don’t think it’s reasonable to plug in the CFG-inference x0 into the formula and define it as "training loss". CFG is applied during the inference time, and the training loss has no relation to the inference. If you want to claim that CFG will make the generation distribution even farther from the original distribution, this is definitely true and is common sense in the diffusion model field, since even the training process of CFG itself is not aimed to minimize ||\hat_{x}_0, x_0||.
* Some prior work has already shown similar insights. For example, [1] shows that CFG would cause memorization if we apply CFG in an early denoising stage. [2] also has similar conclusions. The insight behind this is somewhat intuitive: since the conditional diffusion memorizes the sample, stronger guidance would strengthen the memorization.
* The finding that memorization occurs at a very early stage of denoising is also discussed in some prior work [3].

[1]: Classifier-Free Guidance inside the Attraction Basin May Cause Memorization.
[2]: Feedback Guidance of Diffusion Models.
[3]: Exploring Local Memorization in Diffusion Models via Bright Ending Attention.

**Questions:**

Please refer to previous sections.

**Details Of Ethics Concerns:**

No.

---

> ### Author Response · Authors · 2025-11-21
>
> > **(Weakness 1)** My biggest concern is your claim that overfitting cannot explain memorization alone. For the CFG, if the conditional diffusion does not overfit, then CFG won’t amplify the effect. So, the memorization is still attributed to overfitting. In other words, the overfitting of conditional predictor is the necessary condition, and CFG just gives it a push.
>
> We are not 100\% sure if we understood your concern correctly.
> It would be grateful if you could rephrase your concern, so that we can address your concern accurately.
> Please let us answer based on our understanding at this moment.
> When we mention ``overfitting cannot explain memorization alone``, we meant that for memorization to occur, there is something else (=CFG) that should be applied on top of overfitting, and we didn't mean that either of overfitting and CFG causes memorization.
> Thus, our paper has the exactly same viewpoint as your comment: ``overfitting is the necessary condition``.
>
>
> > **(Weakness 2)** The preliminary is too long, introducing some basic concepts. Put some part in the appendix and only keep the most important part in your main paper, like the noise of the CFG process.
>
> Thank you for the suggestion.
> We agree that the preliminary is basic, but we included it because it is used in later derivations including Equations 10, 13, and 14.
> We are afraid of losing readability if we move the preliminary to the appendix.
> Nevertheless, we can easily do so if necessary.
>
>
> > **(Weakness 3)** In Section 3.2, you use the term "training loss". Are you using $\\lVert \\hat{\\mathbf{x}}\_{0}^{(t)},\\; \\mathbf{x}\_{0} \\rVert$ as the “training loss”? To use a CFG method, we need to train conditional and unconditional diffusion models. Then, the training loss should just be the loss for training the conditional and unconditional models. But I don’t think it’s reasonable to plug in the CFG-inference $\\mathbf{x}\_{0}$ into the formula and define it as "training loss". CFG is applied during the inference time, and the training loss has no relation to the inference. If you want to claim that CFG will make the generation distribution even farther from the original distribution, this is definitely true and is common sense in the diffusion model field, since even the training process of CFG itself is not aimed to minimize $\\lVert \\hat{\\mathbf{x}}\_{0}^{(t)},\\; \\mathbf{x}\_{0} \\rVert$.
>
> Thank you for the comment.
> We would like to make clarification regarding Equation 10.
> The training loss involves the term $\\lVert \\hat{\\mathbf{x}}\_{0}^{(t)},\\; \\mathbf{x} \\rVert$, not $\\lVert \\hat{\\mathbf{x}}\_{0}^{(t)},\\; \\mathbf{x}\_{0} \\rVert$.
> Therefore, during training we do not need the generated image $\\mathbf{x}\_{0}$ but we use the original image $\\mathbf{x}$.
> We hope this clears the concern.
>
>
> > **(Weakness 4)** Some prior work has already shown similar insights. For example, [1] shows that CFG would cause memorization if we apply CFG in an early denoising stage. [2] also has similar conclusions. The insight behind this is somewhat intuitive: since the conditional diffusion memorizes the sample, stronger guidance would strengthen the memorization.\
> [1]: Classifier-Free Guidance inside the Attraction Basin May Cause Memorization.\
> [2]: Feedback Guidance of Diffusion Models.
>
> Thank you for giving us an opportunity to clarify how our work differs from prior research.
> We realize we may not have communicated this distinction clearly enough in the paper, and we apologize for any resulting confusion.
>
> We would like to emphasize that the prior works [1, 2] do not explain *how* CFG contributes to memorization in early denoising.
> [1] reports empirical observations showing that applying CFG before a certain timestep leads to memorization and suggests disabling CFG in early steps as a mitigation, but it does not analyze the underlying mechanism behind this effect.
> We address this gap by showing that early denoising is precisely where classifier-free guidance induces overestimation (Figure 1), linearly amplifying conditional predictions (Figure 6, Equation 13) and injecting excessive information about the training image (Figure 10) (please see **Explanation of Jain et al. (2025).** in Section 4).
> Likewise, [2] proposes adjusting the guidance scale during denoising to improve diversity and reduce memorization (based on findings from earlier studies), but it still does not explain how guidance leads to memorization in early denoising.
>
> Although the idea that ``stronger guidance would strengthen the memorization`` may seem intuitive, we believe that going beyond the intuition and investigating the detailed mechanism theoretically and empirically as in our paper is an important step toward expanding the knowledge in the research community regarding the issue and opening potential opportunities for new ideas based on the investigation results.

---

> ### Author Response · Authors · 2025-11-21
>
> > **(Weakness 5)** The finding that memorization occurs at a very early stage of denoising is also discussed in some prior work [3].\
> [3]: Exploring Local Memorization in Diffusion Models via Bright Ending Attention.
>
> Thank you for pointing the related work.
> [3] (and additionally [Webster 2023]) also noted that memorization occurs at an early stage of denoising.
> Unlike our work, however, these prior works do not explore specifically which mechanism causes emergence of memorization.
>
> Our paper addresses this gap by showing that, under memorization, the model accurately estimates the memorized data (Figure 1(a-c)), and CFG amplifies this estimation (Figure 1(d-f), Figure 6, Equation 13), leading to overestimation from the earliest denoising steps.
> This overestimation quickly removes the initial noise and drives the latent toward the memorized sample (Figure 10).
>
> [Webster 2023] A Reproducible Extraction of Training Images from Diffusion Models, Ryan Webster, arXiv 2023

---

### Official Review · Reviewer_DLXn · 2025-11-01

**Soundness:** 2
**Presentation:** 2
**Contribution:** 3
**Rating:** 4
**Confidence:** 3

**Summary:**

This paper aims to understand how diffusion models, especially for text-conditioned ones, memorise training data. Through the empirical evidence and theoretical derivation, the authors aim to attribute the memorisation to overestimation of memorised data during early denoising. And such an overestimation is driven by the text conditioned noise prediction. Afterwards, the authors demonstrate that such an overestimation leads to memorised images.

**Strengths:**

This paper attempts to deepen the understanding of memorisation of diffusion models, which is an important topic in the community. Through both empirical evidence and theoretical approximation, the authors showed that the text-conditioned noise prediction is the key to inject training image, and leads to memorisation.

**Weaknesses:**

I have the following concerns which require further clarifications from the authors' side. And please correct me if I am wrong.

1. I am not very convinced by the relationship between memorisation and overfitting in this paper. As the training loss essentially should be $||\boldsymbol{\epsilon}-\epsilon\_{\theta}(\boldsymbol{x}_t, \boldsymbol{e}_C)||+\lambda ||\boldsymbol{\epsilon}-\epsilon\_{\theta}(\boldsymbol{x}_t)||$, with $\lambda$ could be a term representing the text prompts are randomly replaced with void during training. Although equation 3,5,6 could be used to transform it into the loss term 10, whether we can still have such a transformation under classifier-free guidance is not clear to me as now the noise prediction has two terms. Can we use classifier-free guidance to predict noise and have the denoising trajectory of $\boldsymbol{x}_t$ and then observe the loss in equation 6 but with text condition?

2. The authors' conclusions significantly rely on some assumptions of approximations, e.g., $||\hat{\boldsymbol{x}}_0^{(t)}-\boldsymbol{x}||\approx 0$, however, at least from Figure 1, this term is about $10^4$, why can we say it approximates 0. Additionally, $\hat{\boldsymbol{x}}_0^{(t)}\approx k\boldsymbol{x}$  is also a very strong assumption.

3. Normally $\boldsymbol{x}_T$ refers to the forward latents, and we assume it follows a gaussian distribution which is already has no information about the training data $\boldsymbol{x}$. During the generation, we will sample a new one based on gaussian distribution, it seems that the theoretical analysis still uses the same one $\boldsymbol{x}_T$ in the denoising process? How to justify this?

4. Given the intuitions in this paper, what are actionable suggestions to mitigate memorisation in diffusion models?

**Questions:**

See the weakness.

---

> ### Author Response · Authors · 2025-11-21
>
> > **(Weakness 1)** I am not very convinced by the relationship between memorisation and overfitting in this paper. As the training loss essentially should be $\\lVert \\boldsymbol{\epsilon} - \\epsilon\_{\theta}(\\mathbf{x}\_{t}, \\mathbf{e}\_{c}) \\rVert + \\lambda \\lVert \\boldsymbol{\epsilon} - \\epsilon\_{\theta}(\\mathbf{x}\_{t}) \\rVert$, with could be a term representing the text prompts are randomly replaced with void during training. Although equation 3,5,6 could be used to transform it into the loss term 10, whether we can still have such a transformation under classifier-free guidance is not clear to me as now the noise prediction has two terms. Can we use classifier-free guidance to predict noise and have the denoising trajectory of $\mathbf{x}_{t}$ and then observe the loss in equation 6 but with text condition?
>
> Thank you for raising this point.
> Intuitively, the training loss may be seen as $\\lVert \\boldsymbol{\epsilon} - \\epsilon\_{\theta}(\\mathbf{x}\_{t}, \\mathbf{e}\_{c}) \\rVert\_2^2 + \\lambda \\lVert \\boldsymbol{\epsilon} - \\epsilon\_{\theta}(\\mathbf{x}\_{t}, \\mathbf{e}\_{\varnothing}) \\rVert\_2^2$ as you indicate.
> However, the actual training procedure is slightly different from that:
> The training is implemented as a stochastic dropout of text conditioning, i.e., while $\\lVert \\boldsymbol{\epsilon} - \\epsilon\_{\theta}(\\mathbf{x}\_{t}, \\mathbf{e}\_{c}) \\rVert\_2^2$ is basically used as the loss, it is probabilistically replaced with $\\lVert \\boldsymbol{\epsilon} - \\epsilon\_{\theta}(\\mathbf{x}\_{t}, \\mathbf{e}\_{\varnothing}) \\rVert\_2^2$.
> This is not the same as using a weighted sum of the two.
> In other words, we do not seek a solution somewhere between $\\lVert \\boldsymbol{\epsilon} - \\epsilon\_{\theta}(\\mathbf{x}\_{t}, \\mathbf{e}\_{c}) \\rVert\_2^2$ and $\\lVert \\boldsymbol{\epsilon} - \\epsilon\_{\theta}(\\mathbf{x}\_{t}, \\mathbf{e}\_{\varnothing}) \\rVert\_2^2$, but $\varnothing$ can be regarded as an additional prompt.
>
> In addition, we would like to clarify that Equation 6 is written as $\mathcal{L} = \\lVert \\boldsymbol{\epsilon} - \\epsilon\_{\theta}(\\mathbf{x}\_{t}) \\rVert\_2^2$ (without prompt embedding $\mathbf{e}\_{c}$) only for the sake of writing, i.e., because the classifier-free guidance (CFG) has not yet been introduced at that point in the paper.
> The actual loss is $\mathcal{L} = \\lVert \\boldsymbol{\epsilon} - \\epsilon\_{\theta}(\\mathbf{x}\_{t}, \\mathbf{e}\_{c}) \\rVert\_2^2$
>  (or $\mathcal{L} = \\lVert \\boldsymbol{\epsilon} - \\epsilon\_{\theta}(\\mathbf{x}\_{t}, \\mathbf{e}\_{\varnothing}) \\rVert\_2^2$ with a small probability), as explicitly written in Equation 15 in Appendix D.1.
> We are sorry if this caused any confusion.
> We added a footnote in Equation 6 and clarified this in the updated paper.
> In conclusion, Equations 3, 5, 6, and 10 are valid under CFG.
>
>
> > **(Weakness 2)** The authors' conclusions significantly rely on some assumptions of approximations, e.g., $\\lVert \\hat{\\mathbf{x}}\_{0}^{(t)} - \\mathbf{x} \\rVert \approx 0$, however, at least from Figure 1, this term is about $10^4$, why can we say it approximates 0. Additionally, $\\hat{\\mathbf{x}}\_{0}^{(t)} \approx k\\,\\mathbf{x}$ is also a very strong assumption.
>
> Thank you for pointing this out.
> $\\lVert \\hat{\\mathbf{x}}\_{0}^{(t)} - \\mathbf{x} \\rVert\_2^2$ in Figure 1 is the \textit{sum} of squared errors over $4 \times 64 \times 64 = 16384$ elements (the latent dimensionality).
> Therefore, an aggregate error of $10^2$ to $10^3$ (where memorized samples lie, as shown by the yellow points in Figure 1(a–c)) actually corresponds to a very small error in each element (in the order of $10^{-3}$ to $10^{-2}$), which is negligible.
> We made this clear in the caption of Figure 1 in the revised paper.
>
> Regarding the expression $\\hat{\\mathbf{x}}\_{0}^{(t)} \approx k\\,\\mathbf{x}$, this is based on the results in Figure 1(d–f), where $\\mathrm{cos}(\\hat{\\mathbf{x}}\_{0}^{(t)}, \\mathbf{x}) \approx 1$ for memorized samples (yellow points).
> Please note that our analysis focuses on the proportionality at $t=T$ rather than all $t$, i.e., $\\hat{\\mathbf{x}}\_{0}^{(T)} \approx g\\,\\mathbf{x}$, indicating overestimation of the memorized sample in the early denoising steps.
> In Section 3.3, we theoretically derive this expression (Equation 13).

---

> ### Author Response · Authors · 2025-11-21
>
> > **(Weakness 3)** Normally $\\mathbf{x}\_{T}$ refers to the forward latents, and we assume it follows a gaussian distribution which is already has no information about the training data $\\mathbf{x}$. During the generation, we will sample a new one based on gaussian distribution, it seems that the theoretical analysis still uses the same one $\\mathbf{x}\_{T}$ in the denoising process? How to justify this?
>
> It seems that our paper was not clear enough in using this notation.
> We apologize for causing confusion.
> To clarify, we do not claim that $\\mathbf{x}\_{T}$ used in the forward and reverse processes are the same instance.
> In the forward process (training), $\\mathbf{x}\_{T}$ is modeled as a Gaussian random variable $\\mathcal{N}(\\mathbf{0}, \\mathbf{I})$, and in the reverse process (inference), $\\mathbf{x}\_{T}$ is a newly sampled noise from $\\mathcal{N}(\\mathbf{0}, \\mathbf{I})$.
> The notation is shared because $\\mathbf{x}\_{T}$ represents an arbitrary random sample from this Gaussian distribution, not a specific instance from training.
> This convention follows the original DDPM paper [1].
>
> [1] Denoising Diffusion Probabilistic Models, Ho et al., NIPS 2020
>
>
> > **(Weakness 4)** Given the intuitions in this paper, what are actionable suggestions to mitigate memorisation in diffusion models?
>
> Thank you for the good suggestion.
> Based on our finding that memorization arises from overestimation of memorized training data during early denoising, we develop a detection and mitigation strategy to suppress such overestimation in early steps.
>
> Specifically, the detection strategy exploits the high correlation shown in Figure 11(c) to detect memorization in real-time;
> we examine $\delta^{(t)} = ( w\_{0}^{(t)} - \sqrt{\bar{\alpha}\_{t}} ) - ( w\_{T}^{(t)} - \sqrt{1 - \bar{\alpha}\_{t}} )$ and when it exceeds 0 (practically we apply a small margin), it is determined that memorization occurs.
> We obtain a Pearson correlation coefficient of $0.85$ between $\delta^{(t)}$ and SSCD score, which suggests it can indeed serve as a real-time detection metric during generation.
> Using this method, we observe that memorization can be detected before $t = 3.31 \pm 3.69$ steps.
>
> Then, the mitigation strategy adjusts the guidance scale to 0 to use unconditional noise prediction $\epsilon\_{\theta}(\mathbf{x}\_{t}, \mathbf{e}\_{\varnothing})$
>  and reduce overestimation due to the specific memorized prompt $c$.
> We find that this detection and mitigation strategy appears successful, as shown in Appendix H of the updated paper.

---

### Official Review · Reviewer_HuT6 · 2025-11-01

**Soundness:** 3
**Presentation:** 2
**Contribution:** 3
**Rating:** 6
**Confidence:** 4

**Summary:**

This paper tackles the memorization issue in diffusion models by providing a principled analysis of why and how diffusion models memorize their training data, which is a well-studied privacy concern in existing diffusion models.
Through theoretical derivations and empirical analyses, the authors show that memorization is driven by early overestimation of training samples during the denoising process, specifically when classifier-free guidance (CFG) amplifies the contribution of the memorized image. This overestimation reduces latent diversity, collapses denoising trajectories, and forces convergence to the memorized sample.
The paper formalizes this process by decomposing intermediate latents and demonstrating near-perfect correlation between deviations from the theoretical denoising schedule and the severity of memorization. The results indicate that memorization is not merely due to overfitting, but to an intrinsic instability in guided denoising dynamics.

**Strengths:**

1. This paper contributes to the reasons for memorization. While prior works have primarily characterized memorization or proposed detection and mitigation strategies, this work focuses on the mechanistic cause underlying the phenomenon. It provides a causal explanation for memorization that unifies and explains previous empirical findings (e.g., from [1] and [2]).
2. The decomposition analysis of intermediate latents, linking theoretical denoising trajectories to empirical memorization, is a novel methodological contribution.
3. The experiments are conducted systematically, that are across multiple diffusion models (Stable Diffusion v1.4, v2.1, and RealisticVision) and sampling schemes.
4. Overall, this work contributes to preserving the training data privacy of diffusion models.

[1] Detecting, explaining, and mitigating memorization in diffusion models. ICLR 2024.

[2] Classifier-free guidance inside the attraction basin may cause memorization. CVPR 2025.

**Weaknesses:**

1. Although the paper provides a precise mechanistic understanding, it stops short of proposing or testing new mitigation strategies derived from its findings (e.g., modified guidance schedules, adaptive early-denoising regularization). A concrete demonstration of how to exploit the discovered mechanism for reducing memorization would make the contribution even stronger, as most related work proposes the corresponding mitigation strategies based on their findings to further prove the insights.
2. Although the classifier-free guidance (CFG) being the key cause of memorization is convincing, the finding has already been well-studied in some early works in the field (e.g., [3] and [4]) that analyzed this to be one of the causes of memorization, and proposed corresponding mitigation strategies for diffusion memorization. It is suggested to elaborate on the difference in this finding to make the contribution clear and unique.
3. Some experiments involve large-scale sampling (e.g., 436 prompts × 50 images × multiple guidance scales). Runtime and resource requirements are not clearly reported, which are recommended to be included for better reproducibility.

[3] Understanding and Mitigating Copying in Diffusion Models. NeurIPS 2023.

[4] Towards Memorization-Free Diffusion Models. CVPR 2024.

**Questions:**

1. Do the authors suggest that similar overestimation phenomena appear in other modalities (audio, video, text-diffusion)?
2. How does the proposed “early overestimation” mechanism behave in unguided or LoRA-fine-tuned models, where classifier-free guidance is weak or absent?
3. Could the correlation between deviation and memorization severity be used as a real-time detection metric during generation?

---

> ### Author Response · Authors · 2025-11-21
>
> > **(Weakness 1)** Although the paper provides a precise mechanistic understanding, it stops short of proposing or testing new mitigation strategies derived from its findings (e.g., modified guidance schedules, adaptive early-denoising regularization). A concrete demonstration of how to exploit the discovered mechanism for reducing memorization would make the contribution even stronger, as most related work proposes the corresponding mitigation strategies based on their findings to further prove the insights.\
> **(Question 3)** Could the correlation between deviation and memorization severity be used as a real-time detection metric during generation?
>
> Thank you for the good suggestion.
> Based on our finding that memorization arises from overestimation of memorized training data during early denoising, we develop a detection and mitigation strategy to suppress such overestimation in early steps.
>
> Specifically, the detection strategy exploits the high correlation shown in Figure 11(c) to detect memorization in real-time;
> we examine $\delta^{(t)} = ( w\_{0}^{(t)} - \sqrt{\bar{\alpha}\_{t}} ) - ( w\_{T}^{(t)} - \sqrt{1 - \bar{\alpha}\_{t}} )$ and when it exceeds 0 (practically we apply a small margin), it is determined that memorization occurs.
> We obtain a Pearson correlation coefficient of $0.85$ between $\delta^{(t)}$ and SSCD score, which suggests it can indeed serve as a real-time detection metric during generation.
> Using this method, we observe that memorization can be detected before $t = 3.31 \pm 3.69$ steps.
>
> Then, the mitigation strategy adjusts the guidance scale to 0 to use unconditional noise prediction $\epsilon\_{\theta}(\mathbf{x}\_{t}, \mathbf{e}\_{\varnothing})$
>  and reduce overestimation due to the specific memorized prompt $c$.
> We find that this detection and mitigation strategy appears successful, as shown in Appendix H of the updated paper.
>
>
> > **(Weakness 2)** Although the classifier-free guidance (CFG) being the key cause of memorization is convincing, the finding has already been well-studied in some early works in the field (e.g., [3] and [4]) that analyzed this to be one of the causes of memorization, and proposed corresponding mitigation strategies for diffusion memorization. It is suggested to elaborate on the difference in this finding to make the contribution clear and unique.\
> [3] Understanding and Mitigating Copying in Diffusion Models. NeurIPS 2023.\
> [4] Towards Memorization-Free Diffusion Models. CVPR 2024.
>
> Thank you for pointing this out.
> We fully agree that clearly articulating our distinguished contribution is important, and we summarize it below.
>
> The key difference between our work and prior studies is that we explain the specific mechanism that text guidance contributes to memorization, whereas earlier works stop at merely conjecturing it as a contributing factor.
>
> For example, [3] acknowledges that text conditioning plays an important role in memorization but does not analyze how it contributes; its role is only noted, not explained.
> Moreover, [3] does not identify CFG as a key cause of memorization, as no analysis of CFG is provided.
>
> Similarly, [4] proposes a guidance strategy, Anti-Memorization Guidance (AMG), aiming at reducing three potential causes of memorization: image duplication, caption duplication, and highly specific prompts.
> However, the paper does not investigate how text guidance contributes to memorization, but focuses on proposing a mitigation strategy without analyzing how memorization is triggered or becomes severe.
>
> We appreciate your suggestion and added this in Section 4 of the updated paper.
>
>
> > **(Weakness 3)** Some experiments involve large-scale sampling (e.g., 436 prompts × 50 images × multiple guidance scales). Runtime and resource requirements are not clearly reported, which are recommended to be included for better reproducibility.
>
> Thank you for the good suggestion.
> As described in Section 3.1, we use 8× NVIDIA GeForce RTX 4090 GPUs.
> With data-distributed parallel (DDP) computing, logging intermediate latents and noise predictions at every denoising step for all models and schedulers used in this paper can be completed in under a day and requires roughly 6 TB of storage.
> We added this in Appendix G of the updated paper.
>
>
> > **(Question 1)** Do the authors suggest that similar overestimation phenomena appear in other modalities (audio, video, text-diffusion)?\
> **(Question 2)** How does the proposed “early overestimation” mechanism behave in unguided or LoRA-fine-tuned models, where classifier-free guidance is weak or absent?
>
> Thank you for your thoughtful questions.
> They raise important and interesting directions for further exploration, but they go beyond the scope of what we can reasonably address within a single paper.
> We mention these as future directions in Appendix I.

---

### Author Response · Authors · 2025-11-21

We sincerely thank all reviewers for their thoughtful feedback.
Their comments helped us strengthen the clarity and enhance the completeness of our paper.
During the rebuttal, we have updated our paper accordingly, which are highlighted in blue:

- p2: Added a footnote to Equation 6 to reduce confusion regarding the training loss expression.
- p3: Added explanation of why the values of $\lVert \hat{\mathbf{x}}_{0}^{(t)} - \mathbf{x} \rVert_2^2$ are negligible under memorization in Figure 1.
- p9: Added a paragraph in the Related Work section to clarify the distinctions between our work and prior studies connecting text guidance and memorization.
- p19: Included additional experiment details, such as runtime and storage for reproducibility.
- p19: Introduced a simple detection and mitigation strategy motivated by our findings, providing further insight and strengthening our contributions.
- p19: Added a Future Work section to outline potential directions for extending this research.

Beyond these revisions, we provide comprehensive, point-by-point responses to each reviewer.
We hope our responses sufficiently address all raised concerns.

---

### Meta-Review · Area_Chair_nZ63 · 2025-12-29

**Summary:**

This paper provides an empirical study of the memorisation issue in diffusion models by investigating the dynamics of diffusion and the denoising process, with observations regarding training loss, conditional prediction of memorised prompts, and intermediate latents.

Reviewers commented positively on the important topic being addressed, the decomposition analysis of intermediate latents is interesting, the experiments are systematic, and the visualisation is clear.

Reviewers commented negatively on the lack of a new mitigation method, the formulation and interpretations of the theoretical analysis, the identified reason of memorisation has been studied before, and other issues.

Overall, combining the recommendations from all reviewers, the weaknesses outweigh the strengths.

**Reviewer Concerns:**

(Weaknesses are indexed using reviewers' original ordering)

For Reviewer HuT6, W1 has been responded to, with some preliminary evidence. W2 has been responded to and acknowledged. W3 has been addressed.

For Reviewer DLXn, W1 and W2 have been addressed. W3 has been responded to. W4 has been responded to, with some preliminary evidence.

For Reviewer Xz4q, W1 has not been resolved due to misunderstanding. W2 is minor and has been addressed. W3 has been addressed. W4 and W5 have been responded.

For Reviewer B5KX, W1 has been responded but the concern has not been fully resolved.

**Reviewer Scores:**

For Reviewer HuT6, the score is likely to be the same or slightly increased.

For Reviewer DLXn, the score is likely to be the same or slightly increased.

For Reviewer Xz4q, the score is likely to be the same.

For Reviewer B5KX, the score is likely to be the same or slightly decreased.

---

### Decision · Program_Chairs · 2026-01-26

Reject